# Dynamic 3D imaging of cerebral blood flow in awake mice using self-supervised-learning-enhanced optical coherence Doppler tomography

Yingtian Pan[1✉], Kicheon Park[1], Jiaxiang Ren[2], Nora D. Volkow[3], Haibin Ling[2], Alan P. Koretsky[4] & Congwu Du[1]

Cerebral blood flow (CBF) is widely used to assess brain function. However, most preclinical CBF studies have been performed under anesthesia, which confounds findings. High spatiotemporal-resolution CBF imaging of awake animals is challenging due to motion artifacts and background noise, particularly for Doppler-based flow imaging. Here, we report ultrahigh-resolution optical coherence Doppler tomography (µODT) for 3D imaging of CBF velocity (CBFv) dynamics in awake mice by developing self-supervised deep-learning for effective image denoising and motion-artifact removal. We compare cortical CBFv in awake vs. anesthetized mice and their dynamic responses in arteriolar, venular and capillary networks to acute cocaine (1 mg/kg, *i.v.*), a highly addictive drug associated with neurovascular toxicity. Compared with awake, isoflurane (2-2.5%) induces vasodilation and increases CBFv within 2-4 min, whereas dexmedetomidine (0.025 mg/kg, *i.p.*) does not change vessel diameters nor flow. Acute cocaine decreases CBFv to the same extent in dexmedetomidine and awake states, whereas decreases are larger under isoflurane, suggesting that isoflurane-induced vasodilation might have facilitated detection of cocaine-induced vasoconstriction. Awake mice after chronic cocaine show severe vasoconstriction, CBFv decreases and vascular adaptations with extended diving arteriolar/venular vessels that prioritize blood supply to deeper cortical capillaries. The 3D imaging platform we present provides a powerful tool to study dynamic changes in vessel diameters and morphology alongside CBFv networks in the brain of awake animals that can advance our understanding of the effects of drugs and disease conditions (ischemia, tumors, wound healing).

[1] Department of Biomedical Engineering, Stony Brook University, Stony Brook, NY 11794, USA. [2] Department of Computer Science, Stony Brook University, Stony Brook, NY 11794, USA. [3] National Institute on Alcohol Abuse and Alcoholism, National Institutes of Health, Bethesda, MD 20857, USA. [4] National Institute of Neurological Disorders and Stroke, National Institutes of Health, Bethesda, MD 20892, USA. ✉email: yingtian.pan@stonybrook.edu

Cerebral blood flow (CBF) is crucial for maintaining the energy supply needed to support synaptic activity through neurovascular coupling. Therefore, CBF along with other hemodynamic measures has been used to link the blood oxygenation level–dependent (BOLD) signals of functional MRI (fMRI) to cellular-level activity from neurons and astrocytes[1,2]. However, current techniques for in vivo cerebrovascular imaging of experimental animals are hindered mainly by the trade-off between imaging depth and spatiotemporal resolution. These include high-field fMRI with single vessel resolution down to arterioles and venules and fast temporal resolution to resolve cerebral blood volume (CBV) and BOLD changes elicited by brain activation[3], microbubble-based ultrasound microscopy for transcranial deep vascular imaging with close to capillary resolution and fast-tracking of red blood cell velocity ($v_{RBC}$)[4], and long-wavelength near-infrared (NIR-II, e.g., >1 µm) fluorescence imaging for visualization of microvasculature with extended penetration depth beyond 3 mm depending on spatial resolution and sensitivity[5,6]. Photoacoustic microscopy (PAM) allows for 3D label-free microvascular imaging of capillary beds and mapping of the hemoglobin oxygenation states in these vessels in mouse cortex at up to ~0.8 mm of depth[7]. Multiphoton fluorescence microscopy is capable of superb spatial resolution and image contrast to resolve 3D capillary networks at a depth of 1.6 mm in mouse cortex and of measuring $v_{RBC}$ by counting fluorescently-stained RBCs (flux) flowing through a capillary[8–10]. Ultrahigh-resolution optical coherence angiography (µOCA) and Doppler tomography (µODT) have advantages for 3D imaging of microvasculature and CBF velocity (CBFv) networks with capillary resolution, in greater details (e.g., arteriolar, venular and capillary flow networks), and at depths of 1.2–1.6 mm from the surface of the mouse cortex[11–14]. µODT has also demonstrated sensitivity and resolution for capturing the microcirculatory CBFv network response to a laser disruption of an arteriole or a capillary, and for detecting cocaine-induced cortical microischemia, vascular disruption, neoangiogenesis, and adaptation, all enabled both by its relatively large field of view and high spatiotemporal resolution[15–17].

Cocaine misuse increases the risk of life-threatening neurologic complications including strokes, hemorrhages, and transient ischemic attacks. About 25% to 60% of cocaine-induced strokes can be attributed to cerebral vasospasm and ischemia[18–21]. Vasoconstriction seems to play an important role among the factors responsible for ischemia[19,22,23]. Indeed, cerebral vasoconstriction after acute cocaine challenge has been angiographically documented in humans[22,24]. Brain imaging studies have documented marked decreases in cerebral blood flow (CBF) and blood volume (CBV) in cocaine abusers[24,25] and animal brains[26,27]. Despite the advances in knowledge derived from imaging technologies such as PET and MRI regarding the effects of cocaine on brain reward, less is known about the acute and chronic effects of cocaine on cerebrovascular networks in vivo. Our 3D µODT measures intrinsic Doppler effect of moving red blood cells to image CBFv, circumventing the need for contrast agents. This is highly desirable for longitudinal imaging studies of cocaine exposures. Our µODT permits 3D CBFv network imaging across arteries, veins, and capillaries[28] and their responses to brain activations in different cortical layers. 3D µOCA/µODT is: (1) quantitative, (2) label-free, (3) of high-velocity sensitivity (<20 µm/s), (4) of high spatial resolution (<6 µm), and (5) able to cover a relatively large cortical volume (e.g., $3 \times 2.4 \times 1.5$ mm$^3$) rapidly.

However, most preclinical studies of the microvasculature have been conducted under anesthesia, which introduces confounds from the anesthetic agents on cellular (e.g., neuronal, astrocytic) activities and cerebral hemodynamics (e.g., CBF, oxygenation changes)[29–31]. Similarly, most studies of the effect of cocaine on cerebral blood flow in laboratory animals have been performed under anesthesia, which could affect the physiological responses to cocaine[30,32]. Thus, neuroimaging studies have started to image awake animals[2,30,33–35], which is challenging due to motion artifacts that can jeopardize the performance of high-resolution image acquisition, especially for Doppler-based flow imaging techniques such as 3D µOCA/µODT[36–38]. To tackle the challenges, here we optimize a flow imaging platform (e.g., µODT setup, mobile cage, cranial window) that incorporate self-supervised deep-learning methods to effectively reduce motion artifacts from awake-behaving mice, and compare the differences of 3D microvasculature (µOCA) and quantitative CBFv networks (µODT) in the sensorimotor cortex in awake vs. anesthetized states (e.g., isoflurane, dexmedetomidine, ketamine). We apply these techniques to measure differences in cerebrovascular responses to acute cocaine between awake vs anesthetized conditions and to assess the effects of chronic cocaine exposure when measured in awake mice.

## Results

**Enhancing 3D microvasculature and CBFv images in awake animals.** For Doppler-based flow detection modalities, high-resolution imaging of brain microvasculature and CBFv networks is highly sensitive to motion-induced noise and artifacts. Unlike prior in vivo imaging studies on anesthetized animals, 3D µOCA and µODT of awake animals are very challenging. To tackle the challenge, we implemented a strategy combining OCT platform optimization and self-supervised learning for effective denoising and motion-artifact removal. OCT platform optimization includes (1) optimizing A-scan rate and points to prioritize fast imaging rate while maintaining sufficient sensitivity for capillary flow detection, (2) miniaturizing probe to minimize vibration noise, and (3) reducing animal motion using custom short, rigid mounts (e.g., Ti head plate, air-floating carbon cage, animal treadmill pre-training). Figure 1 compares two groups of representative µOCA/µODT images in the sensorimotor cortex of

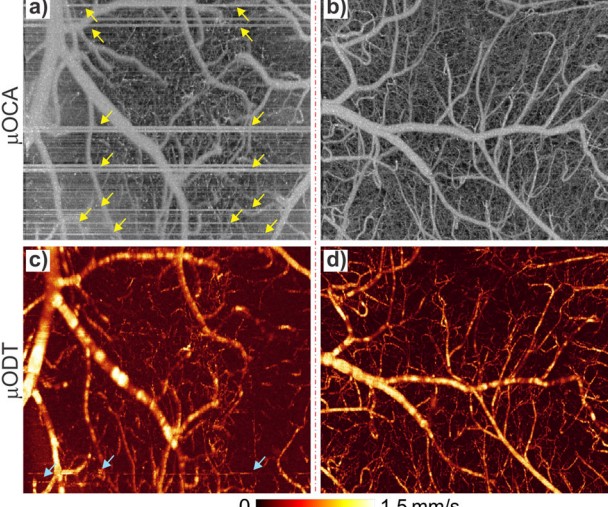

**Fig. 1 Maximum-intensity-projection (MIP) images of 3D µOCA and µODT of awake mice before and after OCT probe redesign and animal treadmill training.** Left panels: µOCA (**a**) and µODT (**c**) images acquired with previously reported OCT scanner; right panels: counterparts acquired with redesigned OCT scanner after animal training (**b**, **d**). Yellow and light blue arrows point to motion artifacts in µOCA and µODT images, respectively. Image size: $2.3 \times 1.2 \times 2$ mm$^3$.

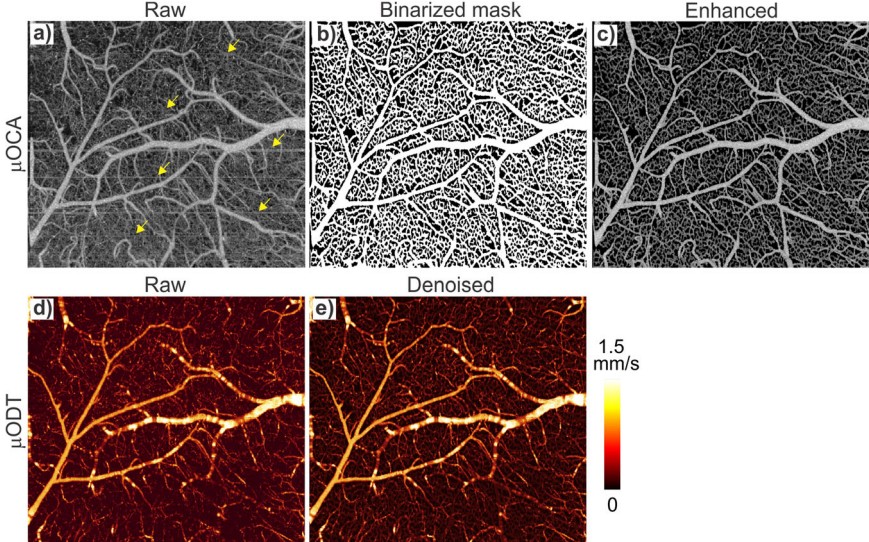

**Fig. 2 Self-supervised learning to minimize motion artifacts and denoise in μOCA (upper panel) and μODT (lower panel) images of awake mouse.**
Yellow arrows (**a**) point to motion artifacts in μOCA that were removed by deep-learning-based image processing (**c**), where the vesselness mask (**b**) was derived from self-supervised learning for motion artifacts removal. Lower panels show the raw μODT image (**d**) was effectively denoised (**e**) by self-supervised learning. Image size 2.3 × 1.2 × 2 mm³.

awake mice before and after OCT scan head redesign and training. The left panels, acquired with our OCT scanner for anesthetized animal studies[13], show severe motion-induced stripes as highlighted by yellow arrows and overall high noise and blurring of microvasculature in μOCA image (a), and motion artifacts (light blue arrows) and excessive background noise in μODT image (c); whereas after OCT scan head redesign and animal treadmill training, the right panels show drastically reduced stripe-like motion artifacts, blurring and background noise in μOCA image (b), and high-fidelity μODT image (d) comparable to those in anesthetized animals[17]. Supplementary videos SV1 and SV2 show the difference in movements before and after treadmill training in a head-restrained mouse. The untrained mouse was very active with frequent movements throughout the session, whereas after training, it became calmer and had substantially less movements than prior to training. Interestingly, even though μODT imaging of capillary CBFv networks is generally more prone to motion-induce phase noise, Fig.1 shows that motion effects (e.g., stripe-like artifacts and noise) were more profound on μOCA than on μODT in awake animals. This is likely due to the shorter time duration between 2 adjacent A-scans to reconstruct μODT (e.g., $\Delta t_{\mu ODT} \approx 2.3$ ms) than between two adjacent B-scans to reconstruct μOCA (e.g., $\Delta t_{\mu OCA} \approx 0.21$ s). The longer duration of μOCA acquisition ($\Delta t_{\mu OCA} \approx 100 \Delta t_{\mu ODT}$) resulted in more severe motion-induced image degradation. Besides, as the μODT images are not angle corrected (i.e., cosine angle between light incidence and flow direction) such as Fig. 1c, d, some branch flows may appear inhomogeneous along the vasculature[28]. More detailed analyses are provided in Supplementary Note S1.

**Deep learning-based processing to reduce motion artifacts in images from awake animals.** As shown in Fig. 1, motion-induced phase noise and artifacts pose a major challenge for awake animal neurovascular imaging, especially for μOCA. In addition to OCT scan head optimization and awake animal training, we implemented various image processing methods, including deep-learning modeling[39–41], and developed a self-supervised learning model to minimize motion artifacts and denoise μOCA and μOCA images (see Methods for detailed frameworks). Figure 2

exemplifies the results of deep-learning-based image processing to reduce motion-induced artifacts (e.g., stripes and the associated blurring of microvessels), in which Fig. 2a is the MIP image projected from the raw μOCA dataset and Fig. 2b is the binarized vesselness mask derived from Fig. 2a by deep-learning segmentation. Figure 2c is the enhanced μOCA image by masking Fig. 2a with Fig. 2b, which shows drastically reduced background noise and clear restoration of the microvascular networks. The effectiveness of our deep-learning framework is evident by the elimination of all the stripe-like artifacts (highlighted with yellow arrows). The lower panels (Fig. 2d, e) compare the μODT images before and after denoising motion-induced phase noise background. The results in Fig. 2 indicate that deep-learning-based image processing minimizes motion-induced noise and artifacts from awake animals, thus enabling more accurate quantitative characterization of microvascular networks and CBFv changes.

**Microvascular networks of mouse sensorimotor cortex in awake vs anesthetized states.** Because of the technical challenges (e.g., motion artifacts), most in vivo brain imaging studies have been performed on anesthetized animals. However, the anesthetic effects confound brain functional responses to electrical or pharmacological stimulations and the associated cerebral vascular changes. Indeed, a visual comparison of the corresponding branch vessels (red/blue bars: arteriolar/venular vessels) in Fig. 3 shows isoflurane (Iso) induced vasodilation. Statistical analyses in Fig. 3e indicate that vessel diameters increased 44.6% ± 4.2% ($p^* < 0.001$, $m = 8$ vessels) in arterioles and 28.2% ± 5.0% ($p^* < 0.001$, $m = 8$) in venules through capillary density remained unchanged (−0.5% ± 0.4%, $p = 0.3$, $m = 8$). In addition to vasodilation, Fig. 4 shows 3D μODT of CBFv networks in awake (a) vs Iso (b) states and their ratio image (c) $\Delta\mu ODT = [\mu ODT(b)-\mu ODT(a)]/\mu ODT(a)$ to illustrate isoflurane-induced CBFv increases, in which red and blue colors refer to CBFv increases and decreases, respectively. To track flow dynamics in the transition from awake to Iso-anesthetized states, a smaller panel of 2.3 × 0.3 × 1.2 mm³ highlighted by a dashed blue box in Fig. 4a was selected to acquire time-lapse 3D μODT stacks (~2 min/cube) as shown in Fig. 4d and the ratio images in Fig. 4e showed overall CBFv increases except in two arterial arcades, i.e., 2 yellow arrows in Fig. 4a. Based

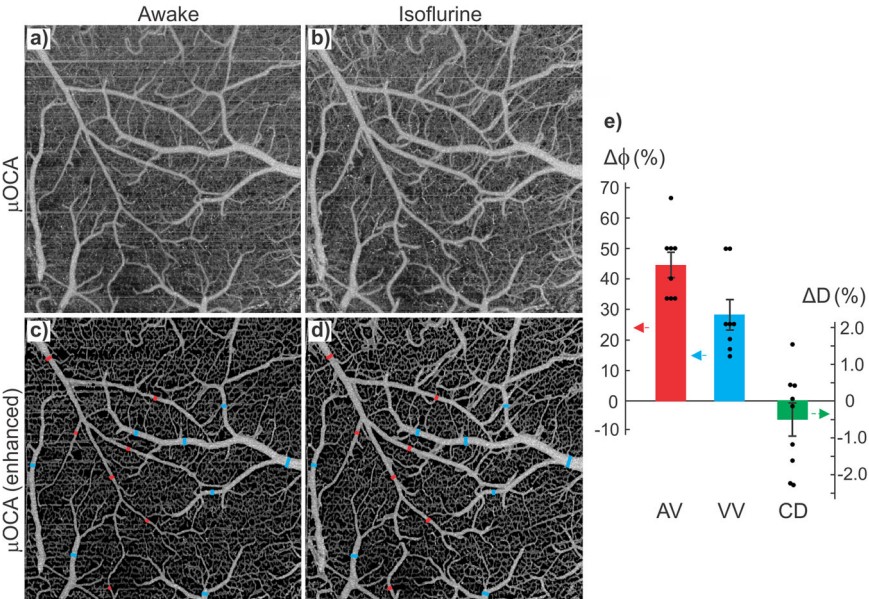

**Fig. 3 3D µOCA images of mouse sensorimotor cortex in awake vs isoflurane (Iso) anesthetized states. a, b** Raw µOCA images of awake vs Iso states; **c, d** their corresponding enhanced µOCA images by deep-learning processing; **e** statistical analyses of isoflurane-induced vasodilation in arteriolar vessels (AV), venular vessels (VV), and capillary density (CD). Image size: $2.3 \times 1.2 \times 2.5$ mm$^3$. Red and blue bars represent arteriolar and venular vessel size changes ($\Delta\phi$), green bar represents capillary density change ($\Delta D$) due to Iso.

on the selected 16 vessels, Fig. 4f plots the flow changes $\Delta CBFv(t)$ in individual vessels (dashed traces) and their average changes, e.g., solid red, blue and green traces for arteriolar, venular and capillary flows. The bold black trace represents the overall flow changes which increased after inhalational isoflurane at $t = 0$ min and plateaued at $t \approx 4$ min (e.g., $50\% \pm 12.9\%$ increase; $p = 0.0003$, $m = 16$). Both arteriolar flow (AF) and venular flow (VF) increased over 70% (AF: $70.9\% \pm 23.1\%$, $p = 0.02$; VF: $72.0\% \pm 20.9\%$, $p = 0.01$); then AF remained relatively stable (e.g., $57.8\% \pm 24.6\%$, $p = 0.05$, $m = 6$), but VF gradually decreased to $22.6\% \pm 10.56\%$, $p = 0.03$, $m = 4$). The capillary flow (CF) changes were more diverse, with increases over 150% and decreases over $-82\%$, yielding a total increase of $29.5\% \pm 22.8\%$ ($p = 0.004$, $m = 6$). Quantification across animals is included in Fig. 4g, indicating that mean $\Delta CBF$ increased $32.75\% \pm 5.65\%$ after iso-flurane induction ($t = 20$–$30$ min, ROIs $= 8$/animal, $n = 7$ mice), which is higher than the normalized baseline ($0.01\% \pm 6.35\%$, $t = -6$–$0$ min; $p^* = 0.02$) in the awake state.

Similarly, we compared the awake vs dexmedetomidine (Dex) anesthetized states. Dex is increasingly favored for animal brain functional studies because of its reported low neurophysiological interference[42,43]. Unlike vasodilation observed with Iso (Fig. 3), 3D µOCA images in Fig. 5 did not reveal vasodilation or vasoconstriction after Dex induction (0.025 mg/kg, i.p.). Quantitative analyses in Fig. 5c indicate that there were no significant changes in vessel sizes, e.g., arteriolar vessels (AV): $-0.3\% \pm 0.7\%$ ($p = 0.67$, $m = 10$), venular vessels (VV): $-0.1\% \pm 0.6\%$ ($p = 0.71$, $m = 16$), and capillary density (CD): $-0.41\% \pm 0.3\%$ ($p = 0.29$, $m = 5$). Figure 6 shows 3D µODT of CBFv networks in awake (a) vs Dex (b) states and their ratio images (c), which revealed minor, scattered flow changes, including both increases and decreases in arterioles and venules under Dex anesthesia. Time-lapse 3D µODT and their ratio images in Fig. 6d, e show the flow changes in the transition from awake to Dex-anesthetized states in the selected ROI (dashed box) in Fig. 6a. Figure 6f plots the relative flow changes in individual vessels (dashed traces, $n = 20$) and the average CBFv changes for arteriolar, venular and capillary compartments. A one-way analysis of variance (ANOVA) that

compared Dex-induced overall flow change (bold black trace) revealed no significant differences over the time $t = 0$ min to 36 min ($F(15, 288) = [1.26]$, $p = 0.23$, $n = 6$ animals). Separate analyses by vessel type showed that arterial flow gradually decreased to $-10.5\% \pm 2.61\%$ ($p = 0.01$, $n = 6$) at $t \geq 9$ min; venular flow decreased to $-14.8\% \pm 3.76\%$ ($p = 0.03$, $n = 5$) at $t \geq 16$ min whereas capillary flow changes were diverse, with increases over 62% and decreases over $-16\%$, yielding a total increase of $6.3\% \pm 10.1\%$ ($p = 0.6$, $n = 8$). Quantification across animals in Fig. 6g revealed no significant $\Delta CBF$ changes between the awake state or baseline ($0.01\% \pm 6.21\%$, $t = -8$–$0$ min) and post Dex anesthesia ($-10.28\% \pm 4.98\%$, $t = 20$–$30$ min, ROIs $= 8$/animal, $n = 3$ animals, $p = 0.24$).

In addition, we imaged the transition from awake to ketamine anesthetized states. Results showed ketamine anesthesia in general caused decreases in regional CBFv. However, its short half-time required multiple injections that led to unstable and heterogenous CBFv changes (Supplementary S3 Fig. s1).

**Cocaine-elicited CBFv network changes in awake vs anesthetized states.** To explore potential confounds from the interaction of different anesthetic agents on cocaine effects on cerebrovascular networks we compared the effects of cocaine (1 mg/kg, i.v.) on the CBFv response in the somatosensory cortex between awake vs Dex or Iso anesthetic states (Fig. 7).

For each group, a full-size 3D µODT image ($2.3 \times 2 \times 1.2$ mm$^3$) was acquired, from which a narrower panel ($2.3 \times 0.3 \times 1.2$ mm$^3$) was selected (Supplementary S4 Fig. S3) to acquire time-lapse dynamic flow network changes (~1.2 min/volume) before and after cocaine. A comparison of cocaine-induced CBFv changes between awake (left panel) and Dex (mid panel) states show a similar pattern of overall flow decreases under Dex anesthesia of $-18.5\% \pm 4.17\%$ and in awake state of $-24.3\% \pm 4.76\%$ ($p = 1$, $n = 5$ animals). Furthermore, the ratio images (**b**) showed that likely due to higher motor activity in sensorimotor cortex in the awake than the anesthetized states, the CBFv changes were more diverse, e.g., exhibiting episodes of short flow bounces after cocaine ($t = 0$ min), especially in venous flows. Self-supervised

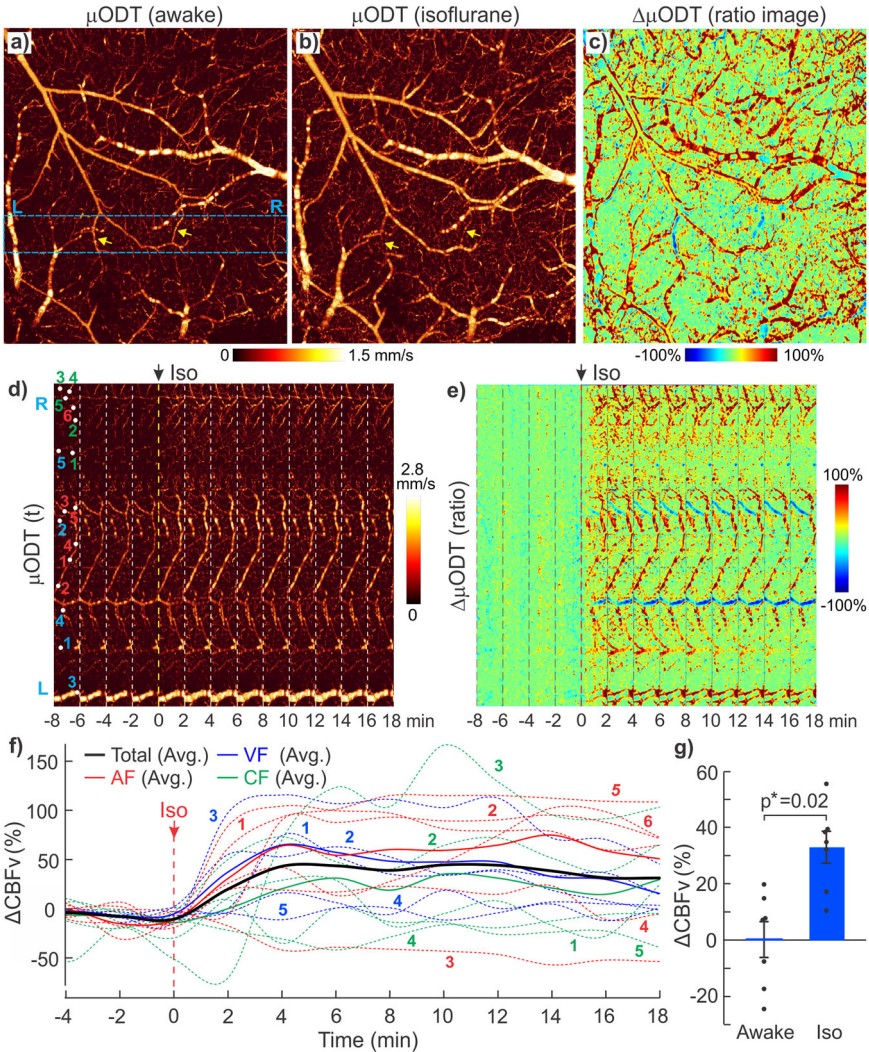

**Fig. 4 3D μODT images of mouse sensorimotor cortex in awake vs Iso-anesthetized states. a**, **b** Raw μODT images of awake vs Iso states and their ratio image. **c** Image size: 2.3 × 2.5 × 1.2 mm³); **d**, **e** time-lapse images μODT(t) and ratio changes ΔμODT from awake to Iso anesthesia (image size: 2.3 × 0.3 × 1.2 mm³); **f** Iso-induced CBFv increases; **g** a comparison of mean ΔCBFv between awake state (t: −8–0 min) and Iso anesthesia (t: 20–30 min) across animals ($p^* = 0.02$, $m = 8$ RoIs/animal, $n = 7$ mice).

learning for motion-artifact tracking in Supplementary S6 Fig. S5 shows a temporal correlation between these transient CBFv episodes and the animal's movements. Interestingly, based on ΔCBFv(t) curves (**c**), the quantified venous rebound amplitudes in awake state (12.01% ± 0.75%, at $t = 2, 8, 20$ min) were significantly higher than in Dex state (4.85% ± 1.64%, at $t = 9, 18, 23, 27$ min; $p = 0.01$). In contrast, the cocaine-induced CBFv decreases with Iso (right panel) were uniform across various vessel compartments (AF, VF, and CF) and larger, i.e., −33.7% ± 2.78% than in the awake state (−24.3% ± 4.76%; $p = 0.04$, $n = 5$) or with Dex (−18.5% ± 4.17%; $p = 0.01$, $n = 5$ animals). It is noteworthy from their full-field basal images (Supplementary S4 Fig. s3) and Fig. 4 that isoflurane dilated vessels, which resulted in ~46% increase in baseline CBFv and facilitated the detection of vasoconstriction triggered by cocaine.

The CBFv network responses to cocaine in the ketamine anesthetized animals showed heterogenous but no significant changes ($p = 0.57$, $m = 14–16$, $n = 3$ animals) (Supplementary S3 Fig. s2).

In addition, we imaged the effects of chronic cocaine on cerebrovascular networks of awake animals ($n = 2$). Cocaine was administered (2 × 1 mg/kg/daily, 2–2.5 h apart, i.v.) roughly

every 3.5 days, i.e., with an accumulated 13 mg/kg of fixed total dose of cocaine over 25–28 days. Figure 8 shows 3D CBFv networks in the sensorimotor cortex between baseline (**a**: day 0) and after chronic cocaine (**b**: day 24). Chronic cocaine-induced vasoconstriction showed an overall reduction in CBFv. Quantification showed overall vasoconstriction Δϕ (**c**) of −22.3% ± 3.1% ($p < 0.001$, $n = 5$) and −25% ± 13.8% for AF ($p < 0.001$, $n = 5$) and −19.8% ± 4.3% for VF ($p = 0.01$, $n = 5$ animals); the detectable capillary flow density ΔD was reduced by −51.7% ± 10% ($p < 0.001$, $n = 5$) based on flow skeleton map analysis[17,44] (Supplementary S5 Fig. s4). The overall CBFv decrease (**d**) was −37.4% ± −4.7% ($p = 0.001$, $n = 5$) among which flow decreases were −25.6% ± 9.3% ($p < 0.002$, $n = 5$), −49.1% ± 27.3% ($p < 0.04$, $n = 5$), and −37.6% ± 5.5% ($p < 0.001$, $n = 5$) for arteriolar, venular, and capillary compartments, respectively. Detailed 3D image analyses in Fig. 9 indicated that the diminished capillary flows after chronic cocaine occurred in upper cortical layers (0–300 μm) whereas deeper cortical layers (300–1000 μm) showed flow increases. Interestingly, unlike CBFv decreases altogether in AF, VF, and CF after acute cocaine as shown in Fig. 9b, e, h, vascular adaptations, e.g., vascular redistribution with extended arteriolar and venular trees in

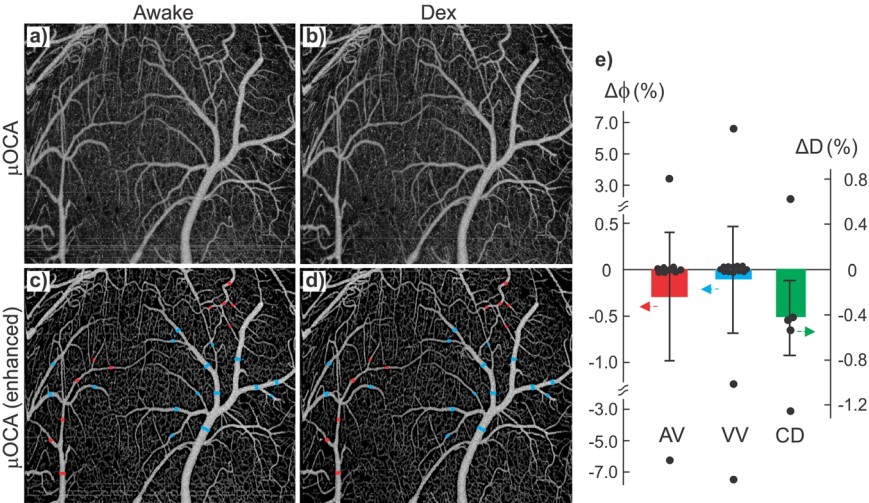

**Fig. 5 3D μOCA images of mouse sensorimotor cortex in awake vs Dex-anesthetized states. a, b** Raw μOCA images of awake vs Dex states; **c, d** Their corresponding enhanced μOCA images by deep-learning processing; **e** statistical analyses of Dex-induced size changes in arteriolar vessels (AV), venular vessels (VV) and capillary density (CD). Image size: 2.3 × 1.2 × 2 mm³. Red and blue bars represent arteriolar and venular vessel size changes (Δφ) respectively; the green bar represents the capillary density change (ΔD) with Dex.

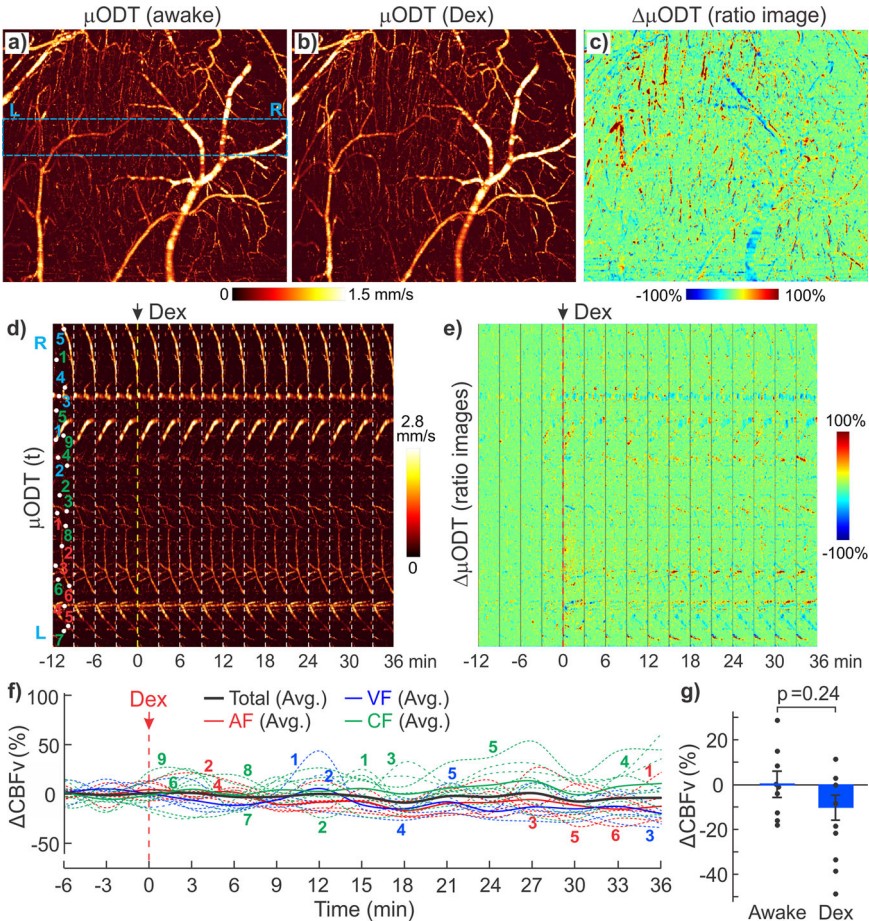

**Fig. 6 3D μODT images of mouse sensorimotor cortex in awake vs Dex-anesthetized states. a, b** Raw μODT images of awake vs Dex states and their ratio image (**c**, image size: 2.3 × 2 × 1.2 mm³); **d, e** time-lapse images μODT(t) and ratio changes ΔμODT from awake to Dex anesthesia (image size: 2.3 × 0.3 × 1.2 mm³); **f** Dex-induced CBFv changes. AF: arterial flow, VF: venular flow, CF: capillary flow; **g** comparison of mean ΔCBFv between awake state ($t$: −8–0 min) and Dex anesthesia ($t$: 20–30 min) across animals ($p = 0.24$, $m = 8$RoIs/animal, $n = 3$).

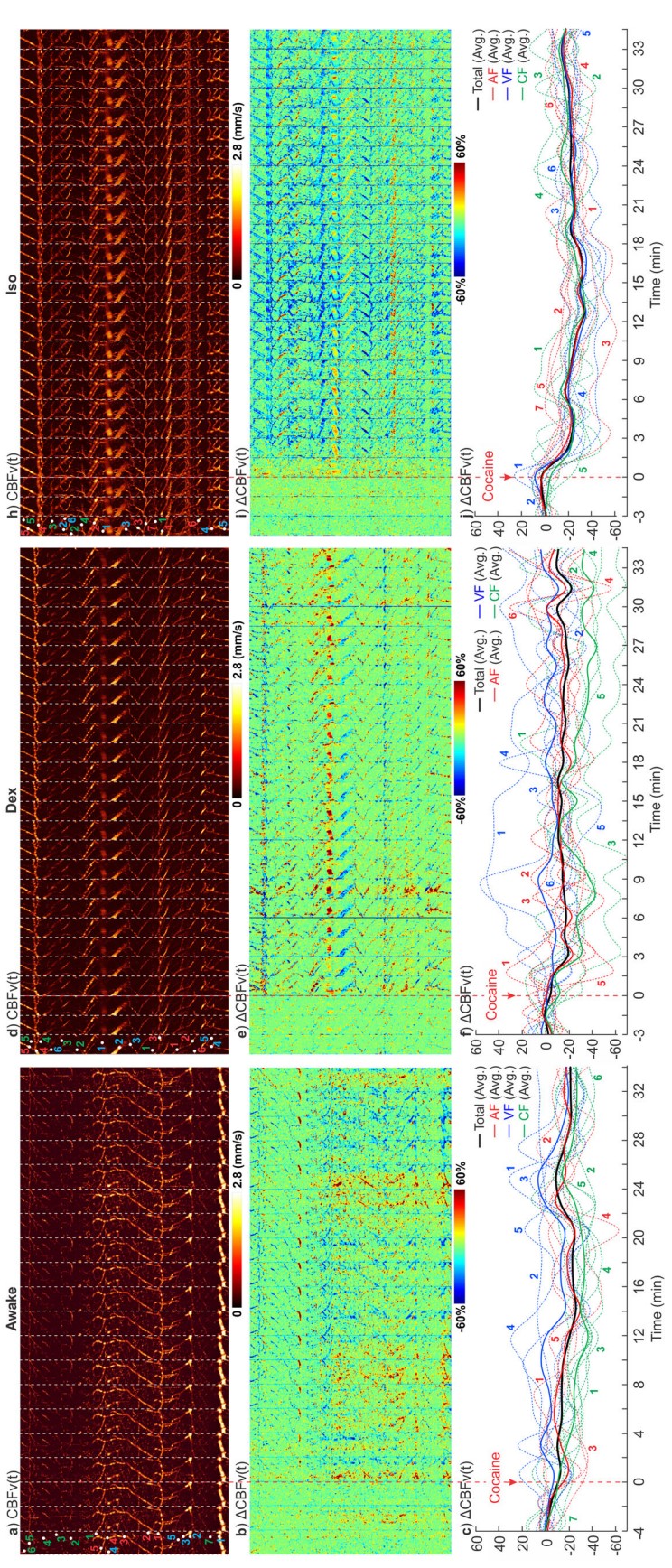

**Fig. 7 Time-lapse spatiotemporal µODT(t) images and their ratio changes in response to an acute cocaine challenge in awake vs anesthetic (Dex, Iso) animal.** Cocaine-induced CBFv changes in the sensorimotor cortex of awake (**a–c**) vs Dex (**d–f**) or Iso (**g–i**) anesthetized mice. Upper panels: time-lapse 3D µODT images of cocaine-induced CBFv changes, µODT(t), color-coded numbers show locations selected to track flow changes in the corresponding lower panels; Mid panels: ratio images of ΔµODT(t), image size: 2.3 × 0.3 × 1.2mm³/panel; Lower panels: cocaine-induced CBFv changes in arteriolar (AF: Red), venular (VF: Blue) and capillary (CF: Green) flow networks. Dashed color traces are individual arteriolar (red), venular (blue), and capillary (green) flow changes with cocaine.

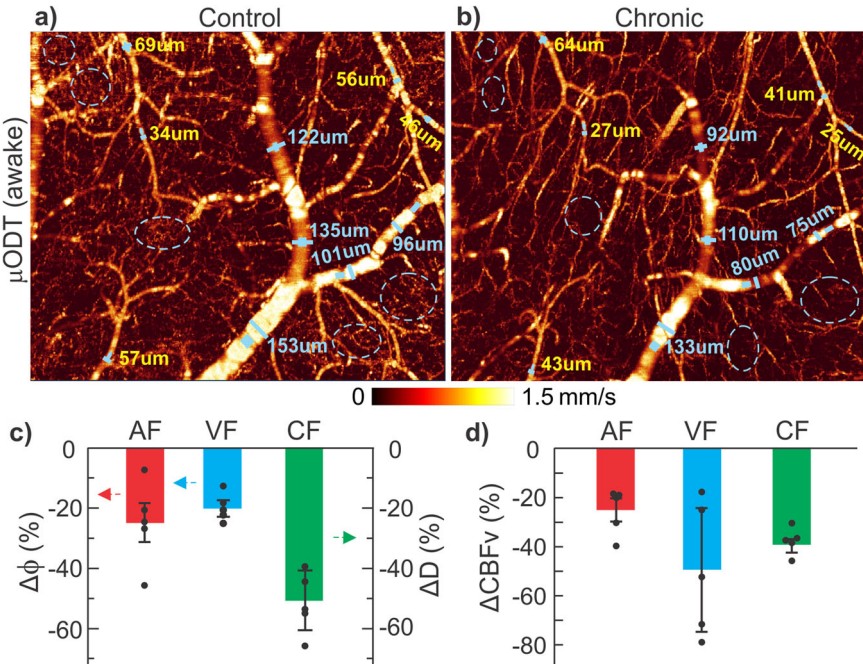

**Fig. 8 Chronic cocaine-induced vasoconstriction and CBFv decreases in the sensorimotor cortex imaged in an awake mouse. a, b** 3D μODT images $(2.3 \times 1.2 \times 2\ mm^3)$ on day 0 (control) and day 24 (chronic cocaine). Solid blue dots and dashed blue circles show ROIs for AF or VF flow and capillary flow quantifications. **c, d** Significant vasoconstriction and CBFv decrease resulting from chronic cocaine exposure in arterial flow (AF: Red), venular flow (VF: Blue), and capillary flow (CF: Green) networks.

Fig. 9c, f, i were observed that appeared to prioritize blood supplies in deeper capillary networks after chronic cocaine as illustrated by the selected 6 diving flows. Additional results are shown in Supplementary S8 Fig. s7.

## Discussion

Doppler-based phase detection is prone to motion noise; therefore, 3D high-resolution CBF imaging of awake animals remains a technical challenge. In this study, we developed deep-learning frameworks to effectively reduce motion artifacts and phase noises in 3D μOCA and μODT images and demonstrated the efficacy of this method for high-resolution imaging of cerebral microvasculature and CBFv networks in the somatosensory cortex of awake-behaving mice. The innovation of this study includes: (1) self-supervised deep-learning method developed and implemented (see Methods), which offers major advantages over prior supervised methods and is suitable for adaptability to generic ODT/OCA systems and different physiological conditions including awake imaging; (2) report on high-fidelity 3D CBFv networks (with μODT) and their dynamic changes in awake animals; (3) comparisons of 3D quantitative tracking of detailed microvascular network dynamics across arterial, venular and capillary flows in awake vs anesthetized animals and in response to acute cocaine; (4) documentation of vascular adaptations following chronic cocaine (e.g., reduced overall flow in the cortex with prioritized capillary flow increases in deep cortex) in awake mice.

As self-supervised learning circumvents the need for large datasets as 'ground truths' for training, it is uniquely suitable to denoise and motion artifact removal in μOCA/μODT images which are usually unpractical to acquire and subject to system changes and variabilities in animal physiology, especially for studies in awake animals. For instance, the results show that although our prior supervised learning worked well for our old OCT platform in anesthetized animal[39], it was unable to restore cortical capillary networks in the awake animal whereas the new

self-supervised learning succeeded (Supplementary S7 Fig. s6). It enables high-fidelity μODT imaging of 3D CBFv network and their dynamic changes in the cortex of an awake animal. This provides a new tool to study brain function based on CBFv network changes in awake animals avoiding the confounds and complications from anesthesia. Although other similar awake animal vascular imaging studies have been reported[45–48], they were applied for enhancing OCA (OCT angiography). Our deep-learning approach is self-supervised not only to remove bulk motion artifacts in μOCA images but also to denoise 3D μODT images in awake animals. In combination with OCT probe optimization and animal treadmill training, it dramatically improved the image fidelity as illustrated in Figs. 1 and 2. It is interesting to note that μOCA was more susceptible to motion artifacts than μODT because of its longer acquisition time used for image reconstruction (Supplementary Note S1). It should also be noted that the detectability of μODT, especially for minute capillary flows, is extremely sensitive to background noise from micromotion of awake animals. Therefore, supervised denoising is necessary to enhance capillary flows (Figs. 2 and 4). Large bulk motion artifacts may also occur during awake animal μODT imaging and can be minimized by our self-supervised-learning-based denoising algorithm (e.g., Fig. 7a). Due to motion artifacts and Doppler phase noise (washout) regardless of using head restraints (e.g., Fig. 1), it is technically very challenging to achieve awake animal flow imaging for a Doppler-based modality. Indeed, our results demonstrate high-fidelity 3D CBFv imaging (not angiography or vasculature imaging) of awake animal using μODT.

The μODT images commonly exhibited multiple bright (high flow) and dark (low flow) spots even along the same vessels. These bright and dark spots were largely repeatable such as those in Fig. 9a, b acquired at different times, suggesting that it is unlikely due to random red blood cells passing through the vessel cross-sections within the μODT B-scans. Instead, as the μODT images presented were not Doppler angle $\theta_z$ corrected

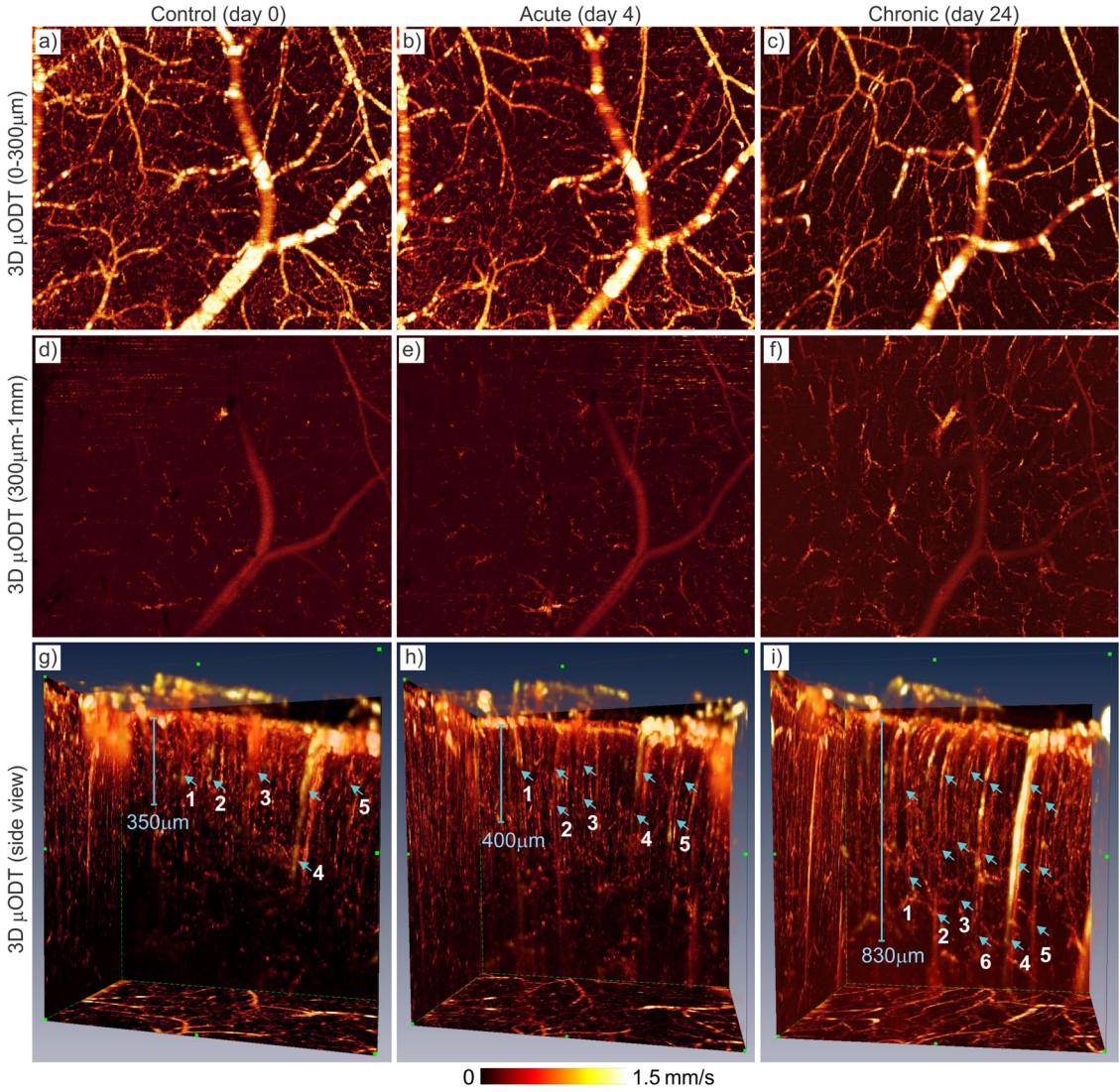

**Fig. 9 Chronic cocaine-induced redistribution of the CBFv networks in the cortex imaged in an awake mouse. a–c** 3D μODT images ($2.3 \times 2 \times 0.3$ mm$^3$) in upper cortex (e.g., L1-L3) on day 0 (control), day 4 (acute cocaine), and day 24 (chronic cocaine), showing reduced capillary CBFv beds and extended AF and VF branches in chronic case (**c**); **d–f** the corresponding 3D μODT images ($2.3 \times 2 \times 0.7$ mm$^3$) in deeper cortex (e.g., L3-L6 and below), showing increased CBFv in AF and VF branches and capillary beds in chronic case (**f**); **g–i** side view of μODT images to illustrate diving arteriolar flows and ascending venular flows extending deeper into lower cortical layers due to capillary flow redistribution resulting from chronic cocaine exposure. Light blue arrows highlight extended depths of the selected six diving arteriolar and ascending venular flows ($i = 1, ..., 6$). Total accumulated cocaine doses: 3 mg/kg for acute and accumulated doses of 13 mg/kg for chronic cases.

(Supplementary Note S1 for details), it is likely the cause of artifacts like bright and dark spots along the flows. We reported 3D Hessian matrix to track vessels angle $\theta_z$, which enables accurate angle correction for absolute CBFv quantification except for near horizontal flows (e.g., $1/\cos(\theta_z) \rightarrow \infty$, when $\theta_z \rightarrow 90^0$)[28].

The effects of commonly used anesthetics (e.g., Iso, Dex) on cerebral vasculature and hemodynamics in the mouse cortex were measured in the transition from awake to anesthetized states. Although the effects of anesthetic agents such as Iso and Dex on the cerebrovasculature are generally known; we showed here that the new advances in μOCA/μODT allowed us to provide a more detailed characterization, including how arterial, venular, and capillary components were affected by Iso and Dex, how fast they reacted, how stable/unstable they became, and how they influenced the vascular and hemodynamic responses to acute cocaine. The new advance also allowed us to assess flow network changes in deep vs. superficial cortical layers with chronic cocaine. These results are relevant because many preclinical brain functional

studies were and may still be performed under anesthesia. In this respect, our findings support the use of Dex as an anesthetic to study cocaine cerebrovascular effects that best mimic the awake state.

It is noteworthy that 3D μODT can bridge the gap between TPM (superior spatial resolution, accurate flow measurement, but limited to only single or very few vessels at a time) and other mesoscopic modalities such as laser speckle imaging that is fast but lacks capillary resolution and depth information. Our studies show the unique value of μODT for studying neurovascular interactions at the high spatiotemporal resolution, and sensitivity and across different cortical layers that avoid biased quantification of capillary flow responses because individual capillary flows are highly diverse; therefore, the ability to track abundant local capillary flow changes is crucial. For example, prior studies of ours and others have demonstrated that Iso anesthesia depressed neuronal activities and dilated cerebral vessels resulting in CBFv increases[19,38–40]. Indeed, here we show dramatic vasodilation in

arteriolar and venular vessels but no changes in capillary density whereas CBFv increases although the changes in individual capillaries are diverse. In contrast, with Dex anesthesia, there was no detectable vasoconstriction and only a minor overall CBFv decrease. In our prior studies, we reported large efforts of Iso but not of Dex on neuronal and astrocyte activities and CBFv[20]. Our current findings provide further evidence that brain functional studies conducted under Dex are less likely to be confounded by anesthetic effects than with isoflurane[31]. We also evaluated ketamine anesthesia (ketamine/xylazine (87.5 mg/kg/12.5 mg/kg cocktail, i.p.), which among several targets acts as N-methyl-D-aspartic acid (NMDA) receptor (NMDAR) antagonist[49,50]. We showed decreased regional CBFv compared with the awake state and observed very unstable cerebrovascular networks, which could reflect the effects of multiple doses of ketamine required to maintain anesthesia (Supplementary S3). Further, inasmuch as ketamine increases dopamine signaling in the brain, this adds another confound when conducting studies with cocaine or with other stimulant drugs.

To assess the confounds from anesthesia when assessing the effects of pharmacological agents in cerebrovascular networks, we selected cocaine for it is not only a very addictive but also a highly neurotoxic drug driven in large part by its cerebrovascular effects that contribute to morbidity and mortality. In fact, the mortality from cocaine misuse has increased dramatically in the US with an estimated 24,538 deaths in 2021. Studying preclinical models of cocaine with in vivo imaging tools is clinically relevant to characterize the mechanisms underlying the vasoconstrictive effects of cocaine and thus help develop interventions to mitigate them.

Acute cocaine reduced overall CBFv in arterial, venular, and capillary networks in awake and Iso- or Dex- anesthetized mice, but the decreases were larger under Iso and similar for Dex and the awake states. However, in the awake state there were more short bounces (episodes) in CBFv, which were likely associated with cocaine-induced motor activities in the sensorimotor cortex of awake animals. This was corroborated by the temporal correlation between these transient CBFv episodes and the animal movements as assessed by the self-supervised learning for motion-artifact tracking algorithms (Supplementary S6 Fig. s5). These results indicate that the magnitude of CBFv decreases after acute cocaine under isoflurane anesthesia prior reports might have been amplified by the vasodilation from the anesthetic and highlights the importance of performing imaging in awake animals.

Finally, we assessed the effects of ketamine on cocaine's cerebrovascular effects. Ketamine, like cocaine, increases dopamine in the brain and has other brain functional effects that could confound studies evaluating the pharmacological effects of cocaine. Specifically, (1) it induces changes in regional CBF, interregional connectivity patterns, and glutamate metabolism[51], (2) it alters the availability of striatal dopamine transporters[52], which are the targets of cocaine's effects, (3) it inhibits neuroendocrine and behavioral consequences of cocaine administration[53], and (4) it supports reinforcement through the disinhibition of dopamine neurons in the ventral tegmental area[54]. Quantitative analysis showed no obvious CBFv change after cocaine under ketamine anesthesia (Supplementary S3).

Prior preclinical studies have documented neurotoxic effects from chronic cocaine including vasoconstriction and ischemia[16,17,27,55]. Most of these imaging studies were conducted under anesthesia (e.g., isoflurane), which likely confounded findings such as from Iso vasodilation that might have undermined long-term vasoconstricting effects of cocaine. Thanks to the chronic cranial window implantation procedure and deep-learning-based motion artifact/noise canceling techniques, we are now able to track detailed cerebrovascular changes resulting from chronic cocaine exposure in awake-behaving animals, thus eliminating anesthesia-induced confounding artifacts. Our findings documented marked cerebrovascular hypofunction in the cortex of awake animals after exposure to chronic cocaine that might underlie the vulnerability to ischemia and stroke with cocaine exposures as reported by our group and others[17]. We also noticed based on the 3D μODT images (Figs. 8, 9) that the diminished capillary CBFv in chronic cocaine animals occurred mostly in upper cortical layers I–III (e.g., 0–300 μm), whereas penetrating arteriolar and venular flows extended to deeper cortical layers (e.g., 300–780 μm), likely to prioritize blood perfusion in these layers presumably to sustain cortical activity in an overall hypoperfused brain. The mechanisms underlying such differential responses to chronic cocaine between cortical layers are unclear but might reflect differences between dopamine and other neurotransmitters and receptors (e.g., noradrenaline) in modulation of flow or activity across the various cortical regions[56–58]. More studies on vasodynamics and the associated neurophysiology is needed to understand the underlying mechanisms of these phenomena, including changes in neuro-astroglio-vascular interactions. It should be noted that we observed severe cerebrovascular hypofunction following repeated relatively small to moderate doses of cocaine (e.g., a total amount of 13 mg/kg of cocaine over 25 days). Therefore, it is reasonable to anticipate that neurotoxicity could be far more devastating if we tested a long-access model of cocaine self-administration for which animals administer up to 45 mg/kg/day of cocaine[55].

A limitation of this study was that the changes in cellular function (e.g., neuronal and astrocyte activity) were not simultaneously recorded, which could have otherwise allowed characterizing differences in sensory- and motor-related neural activity between awake and anesthetized conditions. Therefore, we cannot separate the changes attributed to altered neuronal activity (e.g., sensory- vs. motor-related neural activities) from those reflecting direct cerebrovascular effects for either the anesthetic effects or their interactions with cocaine. Another limitation is that, although the physiological stability of the animals was well maintained during image acquisitions, we did not measure the depth of anesthesia. Besides, unlike vasoconstriction or vasodilation, the effects on capillary networks were quantified as changes in capillary density (e.g., fill factor).

In summary, we developed an innovative self-supervised deep-learning-based μOCA/μODT technique tailored for 3D high-resolution cerebrovascular imaging in awake animals. To demonstrate the potential of this technique for functional brain imaging of awake animals, we applied it to document the potential perturbations of anesthetic confounds on cerebral hemodynamics (CBFv) in response to cocaine. The findings of our study documenting significant interactions between anesthetics and the pharmacological effects of cocaine may be clinically relevant. For example, preclinical and clinical studies reported that the toxic effects of cocaine are accentuated by alcohol[59], which has anesthetic effects[60,61], thus the enhanced toxicity could reflect such interactions. Our findings are also relevant to help interpret preclinical neuroimaging studies of cocaine conducted under anesthesia and inform future studies for selection of an anesthetic agent when the studies cannot be done in awake animals. Our study also contributes to advancing neuroimaging tools including the use of deep-learning-enhanced image-processing techniques that can be used to measure pharmacological effects of drugs in awake-conscious animals. In addition, by applying advanced self-supervised learning, we are able to effectively denoise and minimize motion artifacts, and more importantly, to monitor the motion/movement of awake animal, which is relevant to motor activities. This approach could be used to study chronic cocaine-elicited cerebrovascular

pathology, e.g., transient ischemic attack and the associated paralysis in awake animal model and thus help advance our understanding of how to prevent and treat it[62,63].

## Methods

**Animal preparation**. C57BL mice (Jackson Laboratory) aged 6–8 weeks old were used. A total of 29 mice were used to conduct 53 imaging experiments, in which 13 mice were used in 26 experiments to characterize the flow differences between anesthetized and awake states as a baseline study, and 16 mice were used in 27 experiments to access acute/chronic cocaine effects on cerebral blood flow as an example of the technology for brain functional study. Supplementary S2 Table s1 summarizes the experimental details for animal groups to compare CBFv differences between anesthetized and awake states, and those for studying cocaine's effects on the cortical CBFv networks. All experimental procedures were approved by the Institutional Animal Care and Use Committee at Stony Brook University and conducted according to the National Institutes of Health (NIH) Guidelines for Care and Use of Laboratory Animals.

To implant a cranial window on the cortex of each mouse, after anesthesia with inhalational isoflurane mixed with pure $O_2$ (4% for induction, 1–2% for maintenance), the mouse head was firmly affixed onto a stereotaxic frame with body temperature kept at ~37 °C and a ~3 × 4 mm$^2$ chronic cranial window above the sensorimotor cortex region (A/P −2.0, M/L−2.0) was carefully opened. The

exposed cortical surface was immediately moisturized with 2% agarose gel and affixed tightly with a 160 μm-thick glass coverslip by applying biocompatible cyanocrylic glue first and then dental cement (MIA622, H. E. Parmer Co.) to the edges of the coverslip to secure its attachment with the skull, thus ensuring immobilization of the brain and minimizing motion artifacts during awake imaging. Four microscrews (MX-0090-01SP, Component Supply Co.) were applied to affix a metal head plate to the surrounding skull above the cranial window and secured with dental cement (MIA622, H.E. Parmer Co. Inc.). The wound surrounding the cranial window was sutured, sterilized, and the mouse was given antibiotic and anti-inflammation treatments (if necessary) to ensure long-term optical clearance. The physiological state of the mouse including electrocardiography (ECG), respiration rate, and body temperature was continuously monitored (SA Instruments, NY) during the experiments. All procedures followed the sterilization guidelines for survival surgery.

**Mobile cage training**. A custom air-inflated rodent mobile cage (treadmill) was used to train head-fixed mice to reduce motion artifacts for brain imaging of conscious animals[30]. The cylinder-shaped mobile cage (e.g., φ24 mm × 8 mm) made of rigid, ultralight carbon fiber sheet, was floated ~0.8 mm above an air-inflated table (e.g., φ380 mm) to allow free movement in arbitrary horizontal directions when a mouse attempted to walk or run. The mobile cage created an illusion of free running while keeping the animal's head stationary for motion-free optical imaging of brain function. However, head-fixation induces stress

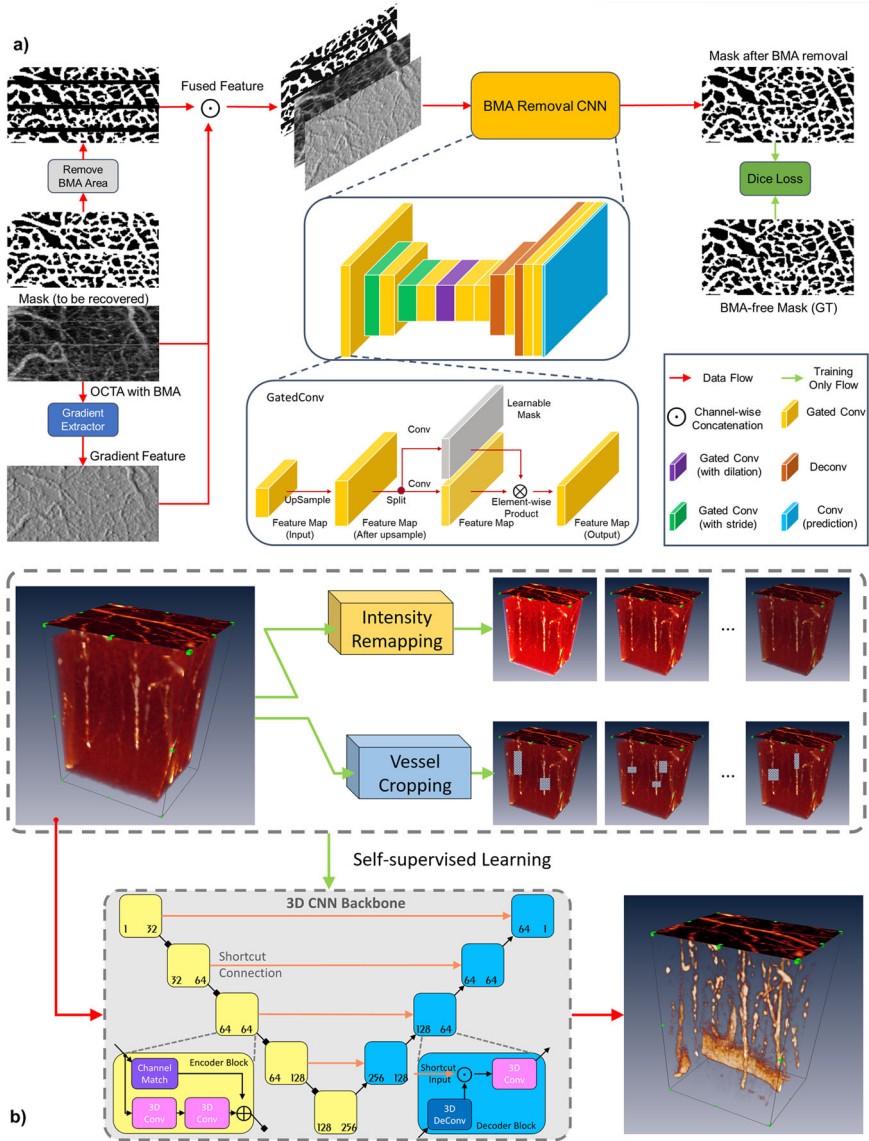

**Fig. 10 Self-supervised deep-learning models to denoise and remove motion artifacts of awake animal in 3D μOCA/μODT images. a** Flow chart of self-supervised deep-learning model to learn motion-induced noise patterns (i.e., green arrows) and to effectively reduce the noises and artifacts (e.g., red arrows). BMA: bulk motion artifacts; CNN: convolutional neural network. **b** Flow chart of self-supervised 3D μODT enhancement model to learn vascular invariant features (e.g., green arrow) for suppressing background noises in the original volume (e.g., red arrow).

that might trigger confounding effects on neurophysiological studies and brain micromotion could in turn compromise optical imaging detection. To habituate the animals to the head-fixed imaging procedure and to minimize motion artifacts, animals were trained under the same condition as those for the awake imaging sessions. Three-day training was adopted with extended multiple training sessions, during which mice were head-fixed while standing on the air-inflated mobile cage with the training time per session progressively increased from 10 min to 30 min and 60 min and the training sessions increased from 1 session on day 1–3 sessions on day 3. The area surrounding the mouse head was draped with black cloth to minimize interference from ambient visual and audio stimulations[64], and signs of stress were monitored by the occurrence of vocalizations and motion during the training sessions[65,66]. The animal protocols for animal preparation, training and in vivo imaging were approved by the Institutional Animal Care and Use Committees of Stony Brook University and followed the National Institutes of Health (NIH) Guideline for Care and Use of Laboratory Animals.

**3D µOCA/µODT of awake-behaving mice**. In vivo 3D imaging of micro-vasculature and cerebral blood flow velocity (CBFv) networks in the sensor-imotor cortex of anesthetized or awake-behaving mice was performed on a custom ultrahigh-resolution optical coherence tomography (µOCT) setup, in which a ultra-broadband light source ($I$ = 8mw; $\lambda$ = 1310 nm, $\Delta\lambda_{FWHM} \approx 200$ nm) illuminated a 2 × 2 wavelength-flattened fiberoptic Michelson inter-ferometer, capable of achieving an axial resolution of 2.5 µm in biological tissue (i.e., coherence length $L_c = 2(ln2)^{1/2}/\pi \cdot \lambda^2/\Delta\lambda_{FWHM}$). Light exiting the sample arm was collimated to ϕ3–4 mm, transversely scanned by a servo minor, and focused through the chronic cranial window on the mouse's sensorimotor cortex by a NIR achromatic doublet (f18 mm/NA 0.25, Edmond Optics), yielding a maximal lateral resolution of 3.2 µm. Light returned from the sample and the reference arms was recombined in the detection fiber, collimated and linearly diffracted by a custom spectrometer, and detected by a 2k-pixel linescan InGaAs camera (2048 R, Sensors Unlimited) at up to 147k A-lines/s. *En-face* maximum-intensity projection (MIP) of CBFv networks was instantaneously reconstructed via graphic processing unit (GPU) boosted custom GUI programming to enable real-time display of flow networks, e.g., at 2 M pixels per cross-sectional or B-scan as fast as 473fps. 3D images of microvasculature (µOCA) and CBFv (µODT) networks in mouse cortex were reconstructed by speckle variance analysis and phase subtraction method, respectively[11,13,15] (Supplementary Note S1 for details).

To minimize motion artifacts critical to 3D µOCA/µODT imaging of awake animals, modifications of the OCT scanner were implemented, including shortening the optical path in the sample arm, use of molded mechano-optics, and design of rigid Ti transitional plate that interconnects the mouse cranial window and the OCT probe. Additionally, image acquisition paradigms were optimized, e.g., increasing repeated B-scans from 4 frames to 14 frames for µOCA and increasing A-scan rate to 8 kHz with reduced A-scan points to 10k for full-field µODT to effectively eliminate motion noise via post-image processing.

**Anesthesia and cocaine administration**. To switch from the awake to the anes-thetized condition during imaging, mice were anesthetized with inhalational 2.0–2.5% isoflurane (Iso) or via intraperitoneal injection of dexmedetomidine (Dex, 0.025 mg/kg, i.p.). The maintenance of anesthesia was confirmed by a lack of pain response and stable breathing rates (Small Animal Instrumentation, Model 1025 L). Cocaine was administered via tail vein injections (1 mg/kg, i.v.) to compare cocaine-induced hemodynamic changes (e.g., vasoconstriction or dilation, CBFv changes) between awake and anesthetized states (e.g., Iso, Dex). For comparison, as an anesthetic agent, ketamine, containing a cocktail of 87.5 mg/kg ketamine and 12.5 mg/kg xylazine, was administered (i.p.) to switch from awake to anesthetized states during imaging, and also to track cocaine-induced CBFv changes in the animal cortex. The detailed experimental procedures are described in Supplementary S3.

**Image processing and motion-artifact removal**. 3D µODT and µOCA were reconstructed by phase subtraction method which derived the flow velocity (Doppler frequency shift v) from the phase difference between 2 adjacent A-scans and by calculating the speckle variance (i.e., normalized standard deviation) among adjacent B-scans, respectively (Supplementary Note S1). µODT was used to discern arterial and venal flows[16]. Briefly, Doppler flow velocity ν is assigned (+/-) as a product of flow direction or phase difference and flow angle $\theta_z$ (i.e., $\cos\theta_z$ "+" for $0^0$-$90^0$, "-" for $90^0$-$180^0$); thereby, by applying Hessian matrix to track $\theta_z$, the flow direction along a vessel can be determined. If the flow direction in a large vessel tree is towards branches (e.g., branching out), it is an artery/arteriole; if the flow directions of the branches merge into larger vessels (e.g., branching in), it is a vein/venule[16].

Artifacts in µOCA induced by bulk motion (e.g., animal movement) tend to increase stripe-like motion artifacts; therefore, suppression of motion artifacts is critical to permit accurate detection of microvasculature, especially for awake animal imaging. Here, this was implemented by a combined approach. As motion artifacts result in substantial decorrelation across multiple B-scans due to

instantaneous drift or jittering, additional B-scans (e.g., $N = 14$) were acquired, among which those most correlated B-scans (e.g., $N' = 6$, $r_{Corr} > 0.9$) were selected to reconstruct the B-scan of µOCA via the speckle variance algorithm for preprocessing to suppress motion artifacts (Supplementary Note S1).

Although B-scan preprocessing was able to correct most motion artifacts resulting from moderate motion, some large and intractable stripe-like noises from severe motion remained, which often spanned several adjacent B-scans (Fig. 2a). To further suppress motion artifacts, a gradient-based method, e.g., optimally oriented flux was employed to produce initial binary segmentation of cerebrovascular networks[44,67]. As the B-scans collapsed by stripe-like noises were discernible, such stripes were first removed and then their regions were recovered by an effective deep-learning-based structure-aware inpainting model, which was designed to fill the vessel masks in the eliminated vesselness regions[40,44]. More specifically, the model integrated the vasculature information in terms of image gradients in the original stripe area to guide the recovery process, which followed the decoder-encoder architecture with GatedConv modules for inpainting[41]. Such structure information brought additional clues that were neglected by existing methods, and thus boosted the robustness of our inpainting model, especially for wide stripes that posed great challenges to these methods. Moreover, the structure information in the clear area was fully utilized to train our model in a self-supervised manner, thus circumventing the need for costly manual annotations. For details, a diagram of the self-supervised learning model is shown in Fig. 10a. After the motion-artifact-free binarized vesselness mask (e.g., Fig. 2b) was generated, it was multiplied with the input MIP µOCA image to provide the enhanced µOCA image to illustrate motion-artifact-free 3D microvascular networks in the mouse brain (e.g., Fig. 2c). The binarized mask Fig. 2b was also used to calculate capillary flow density (CD) as shown in Supplementary S5 Fig. s4.

Meanwhile, a self-supervised learning model was derived to enhance microvascular flows in 3D µODT volumes, which learned the invariant vascular features from Intensity Remapping (IR) and Vessel Cropping (VC) modules. Based on the statistics on data distribution, the IR module generated vessels of different intensities to guide the learning of invariant intensity features. Furthermore, the VC module was used to encourage vasculature connectivity of prediction. To be specific, some segments of vessels were randomly dropped to train the model to learn the vascular features via restoring the removed segments; thus, our model not only improved image contrast but also enhanced vessel connectivity. Compared with other self-supervised methods, our model did not need pairs of clear/noisy or noisy/noisy images, and is, therefore, more practical in the µODT related biomedical applications. UNet-style 3D CNN was used as backbone as illustrated in Fig. 10b. After the training converged, our model converted a noisy µODT volume directly to an effectively enhanced µODT volume (e.g., from Fig. 2d, e. In addition to removing motion artifacts, the self-training method effectively suppressed background noise and improved the image contrast, enhancing weak capillary flow networks in low SNR regions.

To quantify CBFv and compare their differences, roughly 4–6 vessels for each vessel type were selected in each animal to sample their flow rate changes and calculate their means and standard errors, in which the sampling spots (e.g., $i = 1,..., 6$) and the corresponding traces were highlighted in red, blue and green for arterial, venal and capillary flows, respectively. The changes of the microvascular networks (e.g., vasodilation or vasoconstriction) were presented as their relative changes, i.e., $\Delta\phi = (\phi - \phi_0)/\phi_0$, where $\phi_0$ refers to the baseline vessel size. Similarly, the blood flow changes measured by µODT were presented as $\Delta CBFv = (CBFv - CBFv_0)/CBFv_0$, where $CBFv_0$ refers to the corresponding baseline flow rate.

**Statistics and reproducibility**. Statistical tests were performed with SYSTAT Software (Chicago, IL, USA). Differences in CBFv, and vessel diameter between awake and anesthetic states (e.g., ISO, DEX) and between baseline and after cocaine injection were tested by two-tailed $t$ tests or rank sum tests. CBFv changes, vaso-constriction, or dilation were tested for the significant difference using one-way repeated measure ANOVA followed by a post hoc test (Holm–Sidak method). *$p < 0.05$, **$p < 0.01$, ***$p < 0.001$, and NS denotes not significant. All data are presented as mean ± s.t.d. for percent changes.

**Reporting summary**. Further information on research design is available in the Nature Portfolio Reporting Summary linked to this article.

## Data availability
All the source datasets presented in this study are included in the Supplementary Data 1 and 2 files. Additional data, especially for large raw 3D images, are available from the corresponding author upon reasonable request.

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

## Acknowledgements
This research was supported in part by grants 2R01DA029718 (C.D., Y.P.) and 1RF1DA048808 (Y.P., C.D.) from the National Institutes of Health.

## Author contributions
C.D., N.D.V., and Y.P. designed the research. K.P. carried out the in vivo experiments and data analysis; J.R. and H.L. conducted self-supervised-learning-based image processing (K.P. and J.R. contributed equally). Y.P., C.D., N.V.D., and A.K. contributed to data interpretation, result discussions, and manuscript writing.

## Competing interests
The authors declare no competing interests.
