## [Peer Review File · Communications Biology]

Reviewers' comments:

Reviewer #1 (Remarks to the Author):

In this manuscript, Pan and colleagues report ultrahigh-resolution optical coherence doppler tomography (μ ODT) for 3D imaging of CBF velocity in awake animals. The authors apply self-supervised deep-learning framework for effective image denoising and motion-artifact removal, and compare measures of CBFV in awake mice to those anesthetized with isoflurane and dexmedetomidine. All of these results were then compared to those following acute and chronic cocaine exposure. Some major criticisms need to be addressed prior to this manuscript being suitable for publication. Many of these concerns relate to the novelty of the results compared to what has already been reported in other literature, and the consistency of the findings across brain regions or mice. Please see below.

Major

Almost all of the major findings of this paper are known. For example, it has been well documented that isoflurane anesthesia is a potent vasodilator, while the vasoactive effects of dexmedetomidine are less severe. Further, awake imaging is known to be confounded by movement artifacts, so the finding that head restraint and training result in reduce motion artifacts is not novel, and should be left for supplemental (the authors are essentially reporting that they have a good animal preparation, which is great, just not novel). It is also not clear how training the mice to run on the wheel helps remove motion artifacts (top of page 5).

As it is currently written, the motivation for this study is not clearly stated. The Introduction reads like a list of methods previously used to image vasculature. What is novel about the current study? If a major finding/results is that most studies reporting the effects of cocaine are confounded by anesthesia, that should be stated in the abstract and introduction.

The Discussion is almost entirely a rehash of the Results. There is very little "Discussion" or commentary for what the current study adds to the existing literature, what new important information is now accessible with the technology or what applications are in mind for the reported method. If this is a feasibility study or a methods paper, perhaps that should be stated as well.

μ ODT is less affected by the machine learning denoising procedures. Can the authors explain why?

μ ODT results exhibits CBFv within particular vessels. However, the vessels themselves exhibit multiple bright and dark spots. Do the bright spots correspond to individual red blood cells? Please comment. Related, the authors also report CBFv from μ DOT images. Where do these measurements come from in the vessels (bright spots, dark spots, average over a spatial location? Please clarify.

How were arterioles vs venules determined? Are there particular vessels that exhibit the largest changes across all conditions? Further, in Figure 4 and other similar figures, are there equal numbers of vessels for each comparison?

How many mice are used in this study? The consistency of the results across mice is not clear. For example, Fig. 4F show representative results, but how consistent are these results across mice and spatial locations?

There is very little information on the machine learning denoising procedures, yet it is a major component of the results. Please expand the results and discussion to describe this novelty. Similar on page 6, line 106, describe what the "various image processing methods" implemented were to reduce motion artifacts. Did the authors capture rest vs. movement epochs, or quantify motion in any way? If so, it would be interesting to see whether segmenting rest from non rest had the same effect as their ML approaches.

Related to the above - a larger issue with running is also not addressed - running causes local changes in activity within somatomotor cortex, which will result in local changes in blood flow. Please comment. How were these epochs of rest vs. non-rest handled?

Figure 3E – The authors report no change in capillary diameter in awake vs. isoflurane anesthetized mice. This result could simply be use to low CNR/SNR in these vessels, which are very difficult to see in the figure. They authors should report line profiles of baseline diameters in both states, even if supplemental. Similar critique for Fig.5E

Page 13, line 184; Page 18 line 262 – how do the authors know that the change in CBFv are due to metabolic changes?

Fig 9 – Increase contrast in 9D-F. The color scale is driven by what looks to be noise. Panels G-I show depth dependent gradients in flow velocity (I assume, no color bar is provided) with deeper tissue. Are these gradients due to light attenuation with increase depth? Or does flow decrease with increasing depth?

Fig. 9 – The authors only report 1 clear vessel that exhibits “redistribution of CBFv networks”. How consistent were these findings across spatial locations or mice?

Minor

There are several run on sentences throughout the manuscript. Please correct.

General figure comments. Many of the labels within the figure are difficult to read (Fig 4D), or see what type of vessel is being labeled. Please make labels larger and consider choosing fewer “columns” of data. Please arrange Figure 7 to fit on 1 page, and show results side by side instead of the conditions being underneath each other.

Page 18, last paragraph - Please include a citation after the first sentence.

Reviewer #2 (Remarks to the Author):

The authors have examined the blood flow of awake vs. isoflurane and dexmedetomidine anesthetized mice and found that isoflurane induces vasodilation and increases cerebral blood flow.

The manuscript is well organized and the findings are of practical value. My only minor comment is that the authors can talk about the generalizability of their method, i.e., is the motion correction code/program available to the public and whether the coefficients of the neural network will work for experiments done in a different lab.

Reviewer #3 (Remarks to the Author):

Review of manuscript COMMSBIO-22-1480-T titled “Dynamic 3D imaging of cerebral blood flow networks in awake mice using ultrahigh-resolution optical coherence Doppler tomography,” by Pan et al.

This manuscript describes the use of high resolution optical coherence tomography angiography to image flow in the cerebral vasculature in awake mice. The experiments overcome artifacts associated with motion in awake animals by applying AI-based image processing as well as modified optical design.

The title of the paper suggests that the primary novelty is the combination of awake behaving animal

imaging and high resolution OCT angiography. This has been reported previously by Shin et al. Neurophotonics 7.3 (2020) and Rakymzhan et al. Journal of Neuroscience Methods 353 (2021). The particular technological advances that led to the very nice imagery are optics redesign (which is, however, apparently not described here) and the application of an AI approach for data analysis – which the group has reported in their prior publications.

There is a secondary theme of novelty in the paper which is exploring the effects of anesthetics on blood flow in microvasculature. Though not named in the title, this is likely of greater interest to the wider field of in vivo imaging. However, the primary weakness is that pharmacological rationale for that part of the study is not clearly stated. The effects of anesthesia are certainly important, particularly isoflurane which is ubiquitous throughout all animal work, but the choice of dexmedetomidine seems a bit arbitrary. Why not more commonly used anesthetics like ketamine? Also, what is the pharmacological rationale for specifically studying cocaine? The message that I think the authors are trying to convey is that when studying hemodynamic responses to drugs that are not explicitly anesthetics (e.g. here it is cocaine), different forms of anesthesia will affect the drug-associated changes you observe. This is important and has been reported before, albeit possibly not at as high resolution. A challenge however is whether this can be attributed to altered electrical activity or primarily a global flow change. Electrical activity was not monitored and it's not clear if depth of anesthesia and associated decrease in motion was monitored either. From a basic science standpoint, it is therefore hard to formulate a take-home message beyond that indeed anesthesia will affect observed signal sizes (which is relatively common knowledge). If the pharmacological rationale could be re-cast, perhaps focusing on how this solves a challenge in the study of cocaine effects, that might make the article stronger.

Another confound that it is not clear the study has controlled for is that sensory- and motor-related neural activity differs between awake and anesthetized conditions. This is in part because the animals are not actively moving and exploring (on the treadmill, for instance). A concern is whether the neural correlates of sensory and motor changes between awake and anesthetized states influenced the hemodynamic interpretation. Similarly, it is known that spontaneous bursts of neural electrical activity occur during iso anesthesia. Could some of the observed hemodynamic trends be attributed to that?

Minor comments:

Fig. 3c-d – colored bar segments should be more visible – perhaps make them thicker?

Fig. 3e – this is a bit confusing – all of the data points should be shown, i.e. the baseline condition and then the iso condition. Also - the label $\Delta\phi$ should be labeled something more intuitive – what is this metric? How does it apply to CD? Also – it doesn't make visual sense nor communicate much data to the reader to see a ~ 0 $\Delta\phi$ in CD; this is where showing the actual baseline vs. iso condition raw data points could help. Additionally, it should be described in methods how capillary density was calculated.

Fig. 5e – same issues as Fig. 3e. Here, there is basically no information conveyed by the plot. It's not obvious to me what the small numbers punctuating the traces in Figs. 4f, 6f, 7c represent. This should be stated in the figure captions.

Manuscript tracking #: COMMSBIO-22-1480-T

Title: "Dynamic 3D imaging of cerebral blood flow networks in awake mice using ultrahigh-resolution optical coherence Doppler tomography"

Response to the Referee Comments

We would like to express our appreciations to the reviewers for their thoughtful comments and constructive suggestions on the manuscript. We agree with most of the comments and have carefully revised the manuscript accordingly. The revisions are highlighted in *red font* in the revised manuscript. Below are point-by-point responses to the referee's comments.

Reviewer #1:

"In this manuscript, Pan and colleagues report ultrahigh-resolution optical coherence doppler tomography (μ ODT) for 3D imaging of CBF velocity in awake animals. The authors apply self-supervised deep-learning framework for effective image denoising and motion-artifact removal, and compare measures of CBFV in awake mice to those anesthetized with isoflurane and dexmedetomidine. All of these results were then compared to those following acute and chronic cocaine exposure. Some major criticisms need to be addressed prior to this manuscript being suitable for publication. Many of these concerns relate to the novelty of the results compared to what has already been reported in other literature, and the consistency of the findings across brain regions or mice. Please see below."

We agree and have revised the manuscript to clarify the novelty of the methods and the results presented in this work. Please see the reply to Q1 in Editor's Comments above and the changes in the first 2 paragraphs of the Discussion in the revised manuscript.

Major:

1. **a)** *"Almost all of the major findings of this paper are known. For example, it has been well documented that isoflurane anesthesia is a potent vasodilator, while the vasoactive effects of dexmedetomidine are less severe. Further, awake imaging is known to be confounded by movement artifacts, so the finding that head restraint and training result in reduce motion artifacts is not novel, and should be left for supplemental (the authors are essentially reporting that they have a good animal preparation, which is great, just not novel).*

b) *It is also not clear how training the mice to run on the wheel helps remove motion artifacts (top of page 5)."*

a) We agree that the effects of isoflurane (ISO) and dexmedetomidine (DEX) on the cerebrovasculature are generally known; however, we showed here that the new advances in μ OCA/ μ ODT allowed us to provide greater details on the vascular changes and differences between awake vs anesthetized states, including how arterial, venular and capillary components were affected by ISO and DEX, how fast they reacted, how stable/unstable they became, and how the layered flow network changed with acute and chronic cocaine challenges. As reviewer 3 commented these results are important and of practical value, especially because many preclinical brain functional studies were and may still be performed under anesthesia. It is noteworthy that 3D μ ODT can bridge the gap between TPM (superior resolution and very accurate flow measurement but limited to only within single or very few vessels at a time) and other mesoscopic modalities such as laser speckle imaging that is fast but lacks capillary resolution and depth information. Our studies have shown the importance of μ ODT's ability to study neurovascular interactions without the biased quantification when one measures isolated capillary flow responses (individual capillary flows are heterogeneous, which is why quantification of several local capillary flow changes is needed for accurate assessments) and the importance of measuring the cerebral flow network effects across the cortical layers, which requires high resolution and sensitivity. We discuss this in the revised manuscript (pg 19-20)

Due to motion artifacts and Doppler phase noise (washout) even with head restraints (e.g., **Fig.1**), it is technically very challenging to achieve awake animal flow imaging for a Doppler-based modality. To the best of our knowledge, this is the first report of high-fidelity 3D CBFv imaging (not angiography or vasculature imaging) of an awake animal using μ ODT (pg 19-20 in revised manuscript). It was a combined effort of OCT probe redesign and deep learning. Scan head redesign includes (1) optimizing A-scan rate and #s (i.e., to prioritize high A-scan rate while maintaining sufficient sensitivity to detect capillary flow rate), (2) miniaturizing OCT probe to minimize vibration noise, and (3) minimizing mouse motion using short, rigid mounts and Ti rodent head plate and optimizing treadmill design (e.g., air floating pressure, carbon cage, head mount; please see pg 5, para 1-2). The deep-learning diagrams and descriptions are included in Methods of the revised manuscript (pg 30-31, Fig.10).

b) Treadmill training is essential to reduce motion artifacts and it also minimizes confounding physiological responses from being in a new environment/condition. All the data presented except Fig.1 were acquired after 2–3 days of training (2-3 sessions/day, 15-90 min/session). We have included a pair of video clips to compare the difference in the movement of a head-restrained mouse before and after treadmill training. The untrained mouse showed vigorous movement throughout the assessment whereas after training, it was calm and had substantially less movement on the air-inflated treadmill. We have included the video clips in the supplementary movies (**SV1**, **SV2**) of the revised manuscript (pg 6, para 1).

2. "As it is currently written, the motivation for this study is not clearly stated. The Introduction reads like a list of methods previously used to image vasculature. What is novel about the current study? If a major finding/results is that most studies reporting the effects of cocaine are confounded by anesthesia, that should be stated in the abstract and introduction."

The results include both technological advances as well as novel findings regarding cocaine effects, which we included as an example of the value that the technological advances bring to neuropharmacology. We realize that without a specific description of the goals this might have been confusing. To remediate this in the revised manuscript, we modified the abstract and the introduction sections to clarify this (pg 2, ln 7; pg 4-5).

Also, we now specifically clarify the innovation and unique findings from this work: 1) The machine-learning (ML) approach in the current study is self-supervised, which offers major advantages over prior supervised ML methods and is suitable for uses in general ODT/OCA systems and under different physiological conditions (e.g., awake animal); 2) To our knowledge, this is the first report of high-fidelity 3D CBFv networks (with μ ODT) and their dynamic changes in the brain of an awake animal; 3) 3D quantitative tracking of detailed microvascular network dynamics across venular, arterial and capillary flows which we used to compare awake vs anesthetized mice; 4) 3D tracking of acute cocaine induced cerebral blood flow network dynamic changes in awake mice and the vascular adaptations with chronic cocaine across cortical layers (e.g., reduced overall flow in superficial and mid cortex and increases in deep cortical layers) of awake animals. Please see changes in the revised manuscript (pg 18-19; pg 24-25).

3. "The Discussion is almost entirely a rehash of the Results. There is very little "Discussion" or commentary for what the current study adds to the existing literature, what new important information is now accessible with the technology or what applications are in mind for the reported method. If this is a feasibility study or a methods paper, perhaps that should be stated as well."

We agree and have revised the Discussion to point out how this new modality for 3D flow imaging of awake animal could impact current brain imaging studies. We have added the following aspects,

including 1) the innovation of this technique and study; 2) pharmacological rationale for studying cocaine; 3) effects of anesthesia on cocaine's pharmacodynamics in brain, advantages of isoflurane and dexmedetomidine compared to ketamine as aesthetic agents in cocaine studies, and 4) take-home message from this study and its limitations. Please see the changes in the revised version (pg 18-27).

4. "uODT is less affected by the machine learning denoising procedures. Can the authors explain."

Great question. Since the sensitivity of μ ODT is lower than that of μ OCA (e.g., more sensitive to background phase noise, stage vibration) based on findings from previous anesthetized animal imaging studies, we expected that the deterioration of μ ODT images in awake animals would be more serious. However, results showed that μ OCA was more severely affected by bulk motion artifacts in the awake animal. This phenomenon can be explained by the formulas used to reconstruct μ OCA and μ ODT images. Take phase subtraction method (PSM) for simplicity, the flow rate at $v(x_i, \Delta z)$ of μ ODT can be given as:

$$v_i = \frac{\Delta\phi_i \lambda_0}{4\pi n \tau \cos \theta_z} \quad (r1)$$

where k is wave number, λ_0 is central wavelength of the light source, $n \approx 1.38$ is refractive index of brain tissue, Δz is depth from brain surface, θ_z is incline angle of flow, and $\Delta\phi_i$ is the phase difference:

$$\Delta\phi_i = \varphi[F^{-1}[I_{i+1}(k, \Delta z)] - \varphi[F^{-1}[I_i(k, \Delta z)]] \quad (r2)$$

between 2 adjacent A scans at I_{i+1} or $I(x_{i+1}, \Delta z)$ and I_i or $I(x_i, \Delta z)$ with $\tau = \Delta t$ time interval.

The vasculature of μ OCA can be given as:

$$A(x, z) = \frac{1}{\bar{I}(k, \Delta z)} \frac{1}{N-1} \left[\sum_{j=1}^N [I_j(k, \Delta z) - \bar{I}(k, \Delta z)]^2 \right]^{\frac{1}{2}} \quad (r3)$$

the normalized standard deviation among N B-scans, where N is the B-scan numbers (e.g., $N=4$), and $\bar{I}(k, \Delta z)$ is the mean intensity of the cross-section, i.e.,

$$\bar{I}(k, \Delta z) = \frac{1}{N} \sum_{i=1}^N I_i(k, \Delta z) \quad (r4)$$

If we assume the speed of bulk motion artifacts is v_{BMA} , the resultant displacement $\Delta S_{ODT} \approx v_{BMA} \times \Delta t_A$ is significantly less than that of $\Delta S_{OCA} \approx v_{BMA} \times \Delta t_B$, where Δt_A , Δt_B are the duration between 2 adjacent A-scan in μ ODT and B-scan in μ OCA, respectively. For example, $\Delta t_A \approx 1/6 \times 10^{-3} s$, $\Delta t_B \approx 1/27 s$; if considering 14 A-scan oversampling for μ ODT in Eq.(r1) and $N \approx 6$ B-scans in Eq.(r4) for μ OCA, $\Delta t_A \approx 14/6 \times 10^{-3} s$, $\Delta t_B \approx 6/27 s$. In other words, the resultant of motion artifacts on μ OCA is roughly $\Delta t_B / \Delta t_A \approx 100x$ higher than on μ ODT. Therefore, the bulk motion artifacts of awake animal are more serious in μ OCA than in μ ODT images as we observed and the results of machine learning denoising are more obvious. But it should be noted that the detectability of μ ODT, especially for minute capillary flows, is extremely sensitive to background noise floor, which was deteriorated by micromotion in the awake animal. Therefore, machine learning denoising is necessary for enhancing capillary flows (**Fig.4**). Besides, large bulk motion artifacts may occur during awake animal μ ODT imaging and can be effectively minimized by our self-supervised deep-learning denoising algorithm (e.g., updated ratio image in **Fig.7a-b** and 3D images **Figs.9g-i**). Please see the improvement in **Fig.r2** and **Fig.r5** below. We have discussed it in the revised manuscript (pg 7, para 1; pg 20; Fig.10).

5. "uODT results exhibits CBFv within particular vessels. However, the vessels themselves exhibit multiple bright and dark spots. Do the bright spots correspond to individual red blood cells? Please comment. Related, the authors also report CBFv from uDOT images. Where do these measurements come from in the vessels (bright spots, dark spots, average over a spatial location? Please clarify."

Important questions. We noticed multiple bright (high flow) and dark (low flow) spots in μ ODT images, even along a single vessel when we started doing brain imaging studies. First, we thought they were caused by individual red blood cells (RBCs) passing through the vessel cross sections within a μ ODT B-scan. Then, we found that these bright and dark spots were largely repeatable such as those in **Figs.9(a, b)** acquired 4 days apart, suggesting that it is unlikely that they are due to random RBC passings. By tracking individual flows in 3D, we found that they correlated well with the flow angle θ_z according to Eq.(r1), implying that quantification of CBFv is local vascular angle dependent. Because of this, we applied 3D Hessian matrix to track $\theta_z(x, y, z)$ to correct the angle effect. The method works well except for near horizontal flows (e.g., $1/\cos(\theta_z) \rightarrow \infty$, when $\theta_z \rightarrow 90^\circ$).

According to Eq.(r1), μ ODT measurement can derive absolute CBFv quantification and is the only method that can provide quantitative images of CBFv networks and their changes with capillary resolution if angle effect ($\cos\theta_z$) can be accurately corrected (You, et al, 2017). We discuss this in the revised manuscript (Pg 7, para 1; pg 20, para 2).

6. **a)** "How were arterioles vs venules determined? Are there particular vessels that exhibit the largest changes across all conditions? **b)** Further, in Figure 4 and other similar figures, are there equal numbers of vessels for each comparison? **c)** How many mice are used in this study? The consistency of the results across mice is not clear. For example, Fig.4F show representative results, but how consistent are these results across mice and spatial locations "

a) In this study, we used 3D μ ODT to separate venules and arterioles. The detailed procedures were described in our previous publications (Ren, et al, 2012). Briefly, according to Eq.(r1) and Eq.(r2), the measured Doppler flow rates are signed (+/-) depending on (1) flow direction as reflected by the phase difference in Eq.(r2) and (2) the flow angle $\cos\theta_z$ (i.e., "+" for $0-90^\circ$, "-" for $90^\circ-180^\circ$). $\theta_z(x, y, z)$ can be tracked by Hessian matrix, so the flow direction $v(x, y, z)$ along a vessel can be determined. Usually, we started from larger vessels and their immediate branches. If the flow direction in a large vessel tree is towards branches (e.g., branching out), it is arterial/arteriolar by nature. If the flow directions of the branches merge into larger vessels (e.g., branching in), it is venal/venular by nature. This method works well except for arcades, which can be identified because they are arterial vessels.

In the studies (i.e., level of anesthesia, cocaine challenges) presented in our paper, we did not see drastic flow changes different from our previous findings documenting cocaine-induced ischemia following its repeated administration (e.g., across several flows of arterial, venal, or capillary vessel compartments). Under chronic cocaine exposure conditions arcades showed the largest flow changes (e.g., sudden shutdowns). In the current study we document that individual capillary flows can both increase and decrease dramatically but averaging of these changes washes out the heterogeneity of the capillary responses such that the trends appear similar to that in arterial and venal flows.

b) Yes, we tried to select equal numbers of vessels (i.e., $m \approx 4-6$) for each vessel type to quantify CBFv and compare their differences. Take Fig.4 for example. Based on the FOV and the SNR of the small panels for tracking flow dynamics, we chose ~ 16 spots on the vessels to estimate flow changes. Considering that the diversity of venular flows was less than that of arteriolar and especially capillary flows (capillary flows showed most diverse changes), we selected ~ 6 arteriolar flows (labeled as red traces in the dynamic figures), $\sim 4-5$ venular flows (blue traces) and ~ 6 capillary flows (green traces).

c) In this study, a total of 29 mice were used to conduct 53 imaging experiments, in which 13 mice were used in 26 experiments to characterize the flow differences between anesthetized and awake states as a baseline study, and 16 mice were used in 27 experiments to access acute/chronic cocaine effects on CBFv to exemplify the potential of the technology for brain functional studies.

Table r1a below summarizes the experimental details for animal groups to compare CBFv differences between anesthetized and awake states, Table r1b summarizes those for studying cocaine's effects on cortical CBFv, both of which have been included in the Supplemental **S2**.

Also, the results across animals were consistent. Per request, the quantification results across animals are included in the new **Fig.4g** and **Fig.6g**, showing that mean Δ CBF increased $32.75\% \pm 5.65\%$ after isoflurane induction (t=20-30min, ROIs=8/animal, n=7 mice) over the normalized baseline ($0.01\% \pm 6.35\%$, t=-6-0min; $p^*=0.02$) in awake state, whereas there were no significant Δ CBFv changes between the awake state or baseline ($0.01\% \pm 6.21\%$, t=-6-0min) and post Dex anesthesia ($-10.28\% \pm 4.98\%$, t=20-30min, ROIs=8/animal, n=3, $p=0.24$). In addition, the animal numbers for the study of the newly added ketamine anesthesia are included in Table r1a and Table r1b below and in the Supplementary **S2** as well, and the results are included in **Fig.4g** & **Fig.6g** in the revised manuscript (pg 10-11 & pg 12-13; pg 25, para 2).

Table r1a: Experiments and animal groups to compare flows in anesthetized vs awake states

Experiment Descriptions	Total Experiments (Animal #)	Drug Challenge
Awake to Isoflurane anesthesia transition	11 (7)	Inhalational isoflurane (2% in O ₂)
Awake to Dex anesthesia transition	11 (3)	Dexmedetomidine hydrochloride (0.025mg/kg, i.p.)
Awake to ketamine & xylazine anesthesia transition	4 (3)	Ketamine & Xylazine (87.5mg/kg ketamine & 12.5mg/kg xylazine, i.p.)

Table r1b: Experiments and animal groups to study CBFv response to cocaine in anesthetized vs awake states

Experiment Descriptions	Total Experiments (Animal #)	Drug Challenge
Cocaine effect in awake animals	10(7)	Cocaine injection (1mg/kg, i.v.)
Cocaine effect in Isoflurane anesthetized animal	7(4)	
Cocaine effect in Dex anesthetized animals	6(2)	
Cocaine effect in ketamine/xylazine anesthetized animals	4(3)	

7. "There is very little information on the machine learning denoising procedures, yet it is a major component of the results.

a) Please expand the results and discussion to describe this novelty. Similar on page 6, line 106, describe what the “various image processing methods” implemented were to reduce motion artifacts. Did the authors capture rest vs. movement epochs, or quantify motion in any way? If so, it would be interesting to see whether segmenting rest from non-rest had the same effect as their ML approaches. b) Related to the above - a larger issue with running is also not addressed - running causes local changes in activity within somatomotor cortex, which will result in local changes in blood flow. Please comment. How were these epochs of rest vs. non-rest handled? "

a) Very good questions, and we thank the reviewer for pointing out the challenges of this study. We have provided more detailed frameworks about the machine learning procedures for motion artifact removal and denoising in the Method section. Please see the changes in the revised version (**Fig.10**). The major novelty of the machine-learning approach is that it is self-supervised, which offers major advantages over prior supervised machine learning methods for adaptability to generic OCT systems and to the physiological conditions of the animals imaged. This circumvents the need for large training dataset, which is usually unpractical to acquire and subject to system changes and variability of animal's physiology. For instance, the results below in **Fig.r1** show that although our prior supervised ML approach worked well for our old system setup (e.g., OCT scan head) and anesthetized animal, it is unable to restore the majority of capillary networks in the cortex of the awake animal whereas the new self-supervised ML method (**Fig.10**) succeeds. Our prior supervised denoising method (Li et al. 2020; Li, et al., 2021) produces high-contrast image of large, high flows but misses a lot of capillary flows (mid panel). Besides, the original flow rates (color tune) are distorted so additional post-processing is needed to correct the flow rate. Our current method effectively reduces flow noises and retrieves high-contrast weak capillary flows whose flow patterns and flow rates are well preserved compared to the original image (left panel).

Fig.r1 MIP of original μ ODT (left), denoised μ ODT via supervised ML method by A. Li et al (mid), and our self-supervised method.

To our knowledge, this is the first report of high-fidelity 3D CBFv network (with μ ODT) and their dynamic changes in the brain of an awake animal. We believe this will open an avenue for us to study brain function in the context of blood flow changes of awake-conscious brain and avoid the confounds and complications of anesthesia. Our initial machine learning strategy for denoising and bulk motion artifact removal of an awake animal was a supervised approach using images (“ground truth”) of anesthetized animals to train the framework. But we found that the self-supervised approach was more efficient and less system and animal physiology dependent. As reviewer 3 pointed out there have been other similar publications using machine learning, but they are for processing OCA images (angiography). Our ML approach is self-supervised, not only effectively removes bulk motion artifacts and effectively denoises μ OCA, more importantly it effectively denoises and restores time-lapse 3D μ ODT images of the cortex in awake animals (e.g., **Fig.s5**). We discuss this in the revised manuscript (pg 7, para 2; pg 15; pg 18-20) and include the comparison in Supplementary **S7 (Fig.s6)**.

b) Since we imaged the sensorimotor cortex, the CBFv changes are confounded by motor activity (e.g., motion) in the awake animal. Interestingly, animal motion can be tracked and well extracted from our machine-learning framework. The results in **Fig.r5** below illustrate the effectiveness of our motion artifact removal by comparing CBFv(t) before (**a, b**) and after (**c, d**) motion artifact removal that enables tracking of time-lapse animal motion. Compared with the anesthetized state, cocaine decreased CBFv but its increase of animal motion evoked short epochs of flow increases (e.g., $n \approx 10$ in **Fig.r5(c)**), which were associated with movement presumably reflecting cocaine-induced increases in motor activity. This result also implies that machine learning enables us to identify and separate the CBFv changes due to cocaine effects on blood vessels (likely other brain activation) and those due to its enhancement of motor activity. We selected the sensorimotor cortex to study cocaine effects for in our prior work, in which we reported using a μ OCA/ μ ODT-based protocol to image CBFv in isoflurane anesthetized mice, that chronic cocaine induced transient ischemia (TIA) and paralysis/hemiparalysis (motor dysfunction). The new platform reported here now enables us to simultaneously image CBFv in awake mice while monitoring behavioral responses to acute and chronic cocaine, which is research ongoing in our labs. Please see Supplementary **S6** for quantifications.

8. *"Figure 3E – The authors report no change in capillary diameter in awake vs. isoflurane anesthetized mice. This result could simply be use to low CNR/SNR in these vessels, which are very difficult to see in the figure. They authors should report line profiles of baseline diameters in both states, even if supplemental. Similar critique for Fig.5E "*

To save space, we compressed the relative vessel diameter changes ($\Delta\phi/\phi$) of arterioles and venules (AF, VF) and the capillary density change (CD) in the same graph, which caused confusion. We add a right axis ($\Delta D/D$, capillary density) to avoid confusion. Since the method to segment capillary vessels and calculate $\Delta D/D$ was previously published, we now cite a paper (Li, et al, 2017) to reference the detailed image processing and quantification methods. Actually, part of the analysis was illustrated in Supplementary **S5**. We agree that since the spatial resolution of our μ ODT/ μ OCA is $\sim 4\mu\text{m}$ transversely and $\sim 2\mu\text{m}$ vertically, it may not accurately measure the capillary vessel diameters and their minute changes (ΔD). Besides, the μ OCA data were binarized during deep-learning denoising process, so we were unable to provide accurate flow profiles. This is now discussed as a limitation (pg 24, ln 8-9). Please see updated **Fig.3e**, **Fig.5e** and **Fig.8c** for changes.

9. *"Page 13, line 184; Page 18 line 262 – how do the authors know that the change in CBFv are due to metabolic changes?"*

Thanks for the question. In a previous study using multi-channel spectroscopic/fluorescence and laser speckle imaging, we reported overall reductions of neuronal activity (Ca^{2+}), metabolic changes (e.g., ΔHbO_2 , ΔHbR), and CBFv in anesthetized vs awake brain. Since we did not measure $\Delta\text{HbO}_2/\Delta\text{HbR}$ in this study, we have provided a reference (Park et al., 2019). On the other hand, motion tracking (**Fig.r5**) shows a temporal correlation with the 'transient' CBFv bounces (episodes) in the $\Delta\text{CBFv}(t)$ trace. We have modified the statements in the revised version (pg 15, para 2; pg 24, para 1) and included deep learning for motion tracking in the Supplementary **Fig.s5**.

10. **a)** *"Fig 9 – Increase contrast in 9D-F. The color scale is driven by what looks to be noise. Panels G-I show depth dependent gradients in flow velocity (I assume, no color bar is provided) with deeper tissue. Are these gradients due to light attenuation with increase depth? Or does flow decrease with increasing depth?"*

b) *Fig. 9 – The authors only report 1 clear vessel that exhibits "redistribution of CBFv networks". How consistent were these findings across spatial locations or mice?"*

a) Thanks for raising the question. The 3D panel includes 2 overlaid MIP images along z-x and z-y planes, and the noisy appearance is mostly caused by artifacts (vertical strips) underneath large flows,

which is a common problem in ODT and even more serious in OCA images. However, by applying machine learning denoising, we have further reduced the noisy flow background and enhanced the flow clearance to showcase chronic cocaine induced flows diving into deeper cortical layers. In addition, we have included the color bar to show the scale of flow velocity. According to Eq.(r1) and Eq.(r2), the reconstructed CBFv rate is independent of the absolute intensity; however, with the increase of light penetration, the backreflected light intensity attenuates due to more severe multiple scattering, thus the measured flow signal-to-noise ratio decreases accordingly in deeper tissue regions, which affects the detectability of weak flows (e.g., capillary flows). This is now clarified in the revised manuscript.

Fig.r2 3D μ ODT images presented in **Fig.9** (upper panel) and after AI denoising (lower panel). Motion artifacts, e.g., vertical stripes are effectively reduced so that more flows ($i=1, \dots, 6$) can be readily traced to illustrate the redistribution of CBF networks towards deeper cortical layers after chronic cocaine (panel i).

b) We have observed redistribution of CBFv networks towards deeper cortical layers in most chronically cocaine treated animals although this type of *in vivo* imaging study is challenging because it requires good surgical implantation of chronic cranial window that can sustain high optical clearance over 2-4 weeks. Previously, we presented this effect, but it was observed in isoflurane anesthetized mice. After applying ML denoising, the revised **Fig.9(g-i)** readily shows deeper penetration of the diving flows as highlighted by the arrows. **Fig.r2** compares the noisy 3D μ ODT images (upper panel) presented in the original **Fig.9(g-i)** and the updated images after applying deep learning to effectively

denoise and remove motion artifacts such as vertical stripes (lower panel). Six flows are selected to illustrate redistribution of flow network towards deeper cortical layers of chronic cocaine exposed mice. Since μ ODT images 3D CBFv networks over a relatively large field of view, the findings are consistent across spatial locations. In this study, we only presented images in sensorimotor cortex, but we had observed similar features of redistribution of CBFv networks in the prefrontal cortex of chronic cocaine exposed animals (in anesthetized state). In this study, we followed 2 mice over 4 weeks, both of which exhibited redistribution of CBFv networks towards deeper cortical layers when imaged in the awake state (see Supplementary **S8** for mouse 2). In our previous studies in isoflurane anesthetized state, all n=5 animals studied exhibited these findings (You, et al., 2017), therefore the results are consistent across the awake and anesthetized states. Please see the updated **Fig.9(g-i)**, pg 18, para 1 and Supplementary **S8** in the revised manuscript.

Minor:

11. *"There are several run on sentences throughout the manuscript. Please correct."*

Sorry for typos and run-on sentences. We have carefully read the final versions and highlighted the changes in red fonts.

12. *"General figure comments. Many of the labels within the figure are difficult to read (Fig 4D), or see what type of vessel is being labeled. Please make labels larger and consider choosing fewer "columns" of data. Please arrange Figure 7 to fit on 1 page and show results side by side instead of the conditions being underneath each other."*

Thanks for the comments. We have used larger fonts to label the selected vessels with red, blue and green fonts to separate arteriolar, venular and capillary vessels/flows. The columns represent the high temporal resolution of the μ ODT acquisition of the dynamic flow changes, which is challenging due to low SNR to acquire capillary flows and their changes. We have tried to consolidate all 3 panels side by side into 1 page, but the figure was too crowded unless it is presented in landscape format. Please see the updated **Fig.7**.

13. *"Page 18, last paragraph - Please include a citation after the first sentence."*

Thanks, and we have included references (pg 23, para 1).

Reviewer #2:

"The authors have examined the blood flow of awake vs. isoflurane and dexmedetomidine anesthetized mice and found that isoflurane induces vasodilation and increases cerebral blood flow. The manuscript is well organized and the findings are of practical value. My only minor comment is that the authors can talk about the generalizability of their method, i.e., is the motion correction code/program available to the public and whether the coefficients of the neural network will work for experiments done in a different lab."

Thanks for the important comments. A major advantage of our self-supervised machine learning methods is its generalizability compared to prior supervised methods. We will deposit the programs and the coefficients on the website for other labs to use when the manuscript is accepted for publication.

Reviewer #3:

1. "The title of the paper suggests that the primary novelty is the combination of awake behaving animal imaging and high-resolution OCT angiography. This has been reported previously by Shin et al. *Neurophotonics* 7.3 (2020) and Rakymzhanet al. *Journal of Neuroscience Methods* 353 (2021). The particular technological advances that led to the very nice imagery are optics redesign (which is, however, apparently not described here) and the application of an AI approach for data analysis – which the group has reported in their prior publications."

Thanks for raising this issue. Extending current OCT techniques for cerebrovascular imaging to awake behaving animal is a key technological innovation. Compared with recent publications including the 2 referenced papers, our work includes not only ultrahigh-resolution OCT angiography (μ OCA) but also quantitative blood flow networks (μ ODT) and their dynamic changes, including cortical layered-resolved CBFv changes. It is noteworthy that the latter (i.e., the CBFv change measured by μ ODT rather than the vasculature or vascular morphology imaged by μ OCA) is crucial for brain functional studies. We have provided more detailed descriptions of the Methods in the revised manuscript, including moving the self-supervised ML diagrams to the Method (**Fig.10**).

Unlike previous ML approaches (e.g., our prior publications), the ML methods in the current work are self-supervised, which can effectively denoise and remove bulk motion artifacts of awake animals. Importantly, this method is widely applicable to more generic OCT setups and variable physiological conditions of animals being studied (pg 18-20, pg 29-31).

2. "There is a secondary theme of novelty in the paper which is exploring the effects of anesthetics on blood flow in microvasculature. Though not named in the title, this is likely of greater interest to the wider field of *in vivo* imaging. However, the primary weakness is that pharmacological rationale for that part of the study is not clearly stated. **b**) The effects of anesthesia are certainly important, particularly isoflurane which is ubiquitous throughout all animal work, but the choice of dexmedetomidine seems a bit arbitrary. Why not more commonly used anesthetics like ketamine? **a**) Also, what is the pharmacological rationale for specifically studying cocaine? The message that I think the authors are trying to convey is that when studying hemodynamic responses to drugs that are not explicitly anesthetics (e.g. here it is cocaine), different forms of anesthesia will affect the drug-associated changes you observe. This is important and has been reported before, albeit possibly not at as high resolution. **c**) From a basic science standpoint, it is therefore hard to formulate a take-home message beyond that indeed anesthesia will affect observed signal sizes (which is relatively common knowledge). If the pharmacological rationale could be re-cast, perhaps focusing on how this solves a challenge in the study of cocaine effects, that might make the article stronger."

These are important questions, which we will address as follows:

a) The pharmacological rationale for studying cocaine is that it is not only a powerful drug with high addictive potency but also quite neurotoxic, which reflects in part its cerebrovascular effects that contribute significantly to morbidity and mortality. Studying preclinical models of cocaine with *in vivo* imaging tools is clinically relevant for it can help reveal the mechanisms underlying the vasoconstrictive effects of cocaine and thus help develop interventions to mitigate them. As a matter of fact, the mortality from cocaine misuse has increased dramatically in the US to a total of 24,538 deaths in the year 2021 according to recent CDC report. It is likely that the vasoactive effects of cocaine in brain and heart contribute to this mortality. We now strengthen the rational statements for selecting cocaine as an application in the revised version (pg 21-22, pg 23-24).

b) We chose dexmedetomidine (DEX) - an α 2-adrenergic receptor agonist (Montijn et al., 2016), which induces a neural state similar to that of natural sleep (Nelson et al., 2003). Because of this, it has been the choice of anesthetics for most resting-state and neurovascular coupling neuroimaging studies in rodents (Bortel et al., 2020), including our studies on cocaine's effects on spontaneous neuronal firing patterns and vascular activities. Meanwhile, isoflurane (ISO) is a volatile anesthetic

that is advantageous for *in vivo* experimental studies because of its stability to maintain anesthetic depth with a simple inhalational administration. However, ISO causes vasodilation and suppresses central nervous system activities (Rehberg et al., 1996). Taken the pros and cons into consideration, DEX offers the following features: (1) long-term stable physiological conditions (>2hrs) (Pawela et al., 2009; Weber et al., 2006), (2) reproducible BOLD responses to forepaw stimulations similar to those of α -chloralose, which preserves brain activity and neurovascular coupling (Keilholz et al., 2004), and (3) preserves BOLD responses to frequency-dependent stimulation and the strength of resting-state functional connectivity (Jelescu et al., 2014; Magnuson et al., 2014; Pawela et al., 2009). Yes, it remains a question whether or how close the brain function under DEX resembles that of awake animal. Interestingly, our results in this paper show that the responses to acute or chronic cocaine under DEX mimic the responses during the awake state, including that for cerebral hemodynamics. We have discussed it in the revised manuscript (pg 20-21; pg 21, ln 13-16).

Ketamine, on the other hand, has very complex pharmacological properties that among several targets include N-methyl-d-aspartic acid (NMDA) receptor (NMDAR) antagonism (Franks and Lieb, 1994; Zanos et al., 2018). More importantly, like cocaine, ketamine increases dopamine in the brain, which could confound the physiological and pharmacological effects of cocaine. Specifically, (1) it induces changes in regional CBF, interregional connectivity patterns and glutamate metabolism (Bryant et al., 2019), (2) it alters the availability of striatal dopamine transporters (Tsukada et al., 2001), which are the targets of cocaine's effects, (3) it (often used as a mixture for anesthesia) inhibits neuroendocrine and behavioral consequences of cocaine administration (Torres et al., 1994), and (4) it supports reinforcement through the disinhibition of dopamine neurons in the ventral tegmental area (Simmler et al., 2022). Because ketamine's pharmacological effects could confound our experiments, we did not choose it as an anesthetic in our initial study.

Fig.r3 Comparison of 3D μ ODT images of CBFv networks in mouse sensorimotor cortex in awake (a) vs ketamine-anesthetized (b) states, and their ratio image (c), showing unstable CBFv changes under ketamine anesthesia (c). Image size: $2.25 \times 2 \times 1.2 \text{mm}^3$

In response to the reviewer's request, we conducted additional experiments with ketamine as anesthetic. The details of experimental procedures and results are summarized as follows:

- (1). Full-field awake μ OCA/ μ ODT scan: FOV= $2.25 \times 2.0 \times 1.2 \text{mm}^3$;
- (2). Dynamic μ ODT scans in the transition from awake to ketamine states: $2.25 \times 0.3 \times 1.2 \text{mm}^3$ /volume per 1.5-2min over 40min, including ~10min baseline to 10min after *i.p.* injections of anesthetic cocktails, e.g., an initial dose of ketamine (87.5mg/kg) + xylazine (12.5mg/kg), following-up doses of ketamine (29.2mg/kg) + xylazine (4.2mg/kg);
- (3). Full-field ketamine anesthetized μ OCA/ μ ODT scan: FOV= $2.25 \times 2.0 \times 1.2 \text{mm}^3$;

- (4). Dynamic μ ODT scans of cocaine effects: $2.25 \times 0.3 \times 1.2 \text{ mm}^3$ /volume per 1.5-2min over 40min, including ~ 9 -10min baseline to 36min after cocaine injection (1mg/kg, i.v.) during which follow-up i.p. doses of ketamine (29.2mg/kg) + xylazine (4.2mg/kg) were infused;
- (5). Full-field ketamine anesthetized μ OCA/ μ ODT scan after cocaine: $\text{FOV} = 2.25 \times 2.0 \times 1.2 \text{ mm}^3$.

Due to the short half-time (~ 20 -30min) of ketamine/xylazine cocktail for anesthetics (Rosenbaum SB, et al, 2022; Lamont LA, et al., 2000; Veilleux-Lemieux D, et al., 2013), additional 3 injections were given for each experiment after the initial dose to stabilize the animal for experiments. **Fig.r3** shows a pair of representative 3D μ ODT images of the CBFv networks of mouse cortex in awake (a) vs ketamine-anesthetized (b) states. Their ratio image (c) reveals that ketamine-induced CBFv decreased in the first 20min post ketamine injection (blue regions) followed by a partial recovery with

Fig.r4 CBFv responses to cocaine in ketamine-anesthetized animal. **a-b)**: time-lapse images μ ODT(t) before and after cocaine injection (1mg/kg, i.v., at t=0min) and their ratio changes $\Delta\mu$ ODT, image size: $2.25 \times 0.3 \times 1.2 \text{ mm}^3$; **c)** cocaine-induced CBFv changes of individual vessels; **d)** Statistical comparison of CBFv changes before and after cocaine (ROIs=14-16/animal, n=3), showing no significant change ($p=0.57$).

a slight overshooting (yellow and red regions) as a result of short lifetime of ketamine anesthesia. Despite the complications, ketamine anesthesia in general resulted in regional CBFv decreases. Please see pg 13, para 2; pg 28, para 2 for changes in the revised manuscript.

To further track cocaine-induced CBFv dynamic changes in the cortex of ketamine-anesthetized animals, similarly a smaller panel of $2.25 \times 0.3 \times 1.2 \text{ mm}^3$ in **Fig.r3(b)** was selected to acquire time-lapse 3D μ ODT images in **Fig.r4(a)** and their ratio images in **Fig.r4(b)**. **Fig.r4(c)** plots the relative flow changes in individual vessels (dashed traces, $m=16$), showing cocaine-induced inhomogeneous CBFv responses, e.g., increasing and decreasing fluctuations in arteriolar and venular flows, especially in capillaries (green traces). Such temporally inhomogeneous flow responses to cocaine are consistent with their spatially inhomogeneous responses shown in **Fig.r3(c)**. **Fig.r4(d)** summarizes the comparison of CBFv changes in ketamine anesthetized state before and at 20-30min after cocaine across animals (ROIs=14-16/animal, $n=3$), showing no significant difference ($p=0.57$). The inhomogeneous responses to cocaine in the neurovascular network might imply the confounding effects of ketamine on neuronal activities and cerebral hemodynamics. These experimental data are now included in the Supplemental **Fig.s1 and Fig.s2** and discussed in the revised manuscript (pg 22, para 3).

c) Take-home message: We developed an innovative AI-enhanced μ OCA/ μ ODT technique tailored for 3D high-resolution cerebrovascular imaging in awake animals. To demonstrate the potential of this technique for functional brain imaging of awake animals, we applied it to document the potential perturbations of anesthetic confounds on the cerebral hemodynamics (CBFv) in response to cocaine. The differences in cocaine-induced hemodynamic changes within the cortex using different anesthetic agents are likely associated with their anesthetic effects on the cerebral vessels and as well as their effects on neuronal and astrocytic activities. Our findings documenting significant interactions between anesthetics and the pharmacological effects of cocaine are clinically relevant. For example, preclinical and clinical studies have shown that cocaine's toxic effects are significantly accentuated by alcohol (Boag and Havard, 1985) and inasmuch as alcohol possesses anesthetic effects (Busse and Riley, 2003; Goldstein, 1984), this could reflect such interactions. Our results are also relevant to help interpret preclinical neuroimaging studies of cocaine conducted under anesthesia and inform future studies for selection of an anesthetic agent when studies cannot be done in awake animals. Finally, our study contributes to advancing neuroimaging tools including the use of AI-enhanced image processing strategies that can be used to measure pharmacological effects of drugs in non-anesthetized animals. Please see pg 24-25 for changes in revised manuscript.

3. *"A challenge however is whether this can be attributed to altered electrical activity or primarily a global flow change. Electrical activity was not monitored and it's not clear if depth of anesthesia and associated decrease in motion was monitored either.*

In this study, the experimental design was focused on the comparison of CBFv responses to cocaine between awake and anesthetized states. The anesthetic doses of ISO (2-2.5%) and DEX (0.025mg/kg, i.p.) were optimized to stabilize the animal's physiology based on our prior imaging studies (Gu et al., 2018a; Gu et al., 2018b; Park et al., 2019; Park et al., 2021). Though we did not measure the depth of anesthesia, the physiological stability was well maintained during image acquisitions.

In the revised version, we now discuss in the limitation section (pg 24, para 1) that we can not separate changes attributed to altered electrical activity or to global flow changes since we did not monitor them. We also discuss as a limitation that we did not measure the depth of anesthesia. But interestingly, by applying AI denoising for motion artifact removal (see **Fig.r5(c, d)** vs **Fig.r5(a, b)**

below), we were able to monitor the motion/movement of the animal with the 'motion index' to quantify motion artifact severity (**Fig.r5(e)**), which is relevant to motor function (see Supplementary **S6**). Because the animal behavior (e.g., movement) is sensitive to light, imaging studies were performed in dark lighting conditions, so we did not video record the animal movement. In the future, we will install a NIR illumination/camera setup to simultaneously record the movement of awake/conscious animal in subsequent studies.

4. "Another confound that it is not clear the study has controlled for is that sensory- and motor-related neural activity differs between awake and anesthetized conditions. This is in part because the animals are not actively moving and exploring (on the treadmill, for instance). A concern is whether the neural correlates of sensory and motor changes between awake and anesthetized states influenced the hemodynamic interpretation. Similarly, it is known that spontaneous bursts of neural electrical activity occur during iso anesthesia. Could some of the observed hemodynamic trends be attributed to that? "

Fig.r5 Effectiveness of AI denoising to minimize motion artifacts (**a-b**, **c-d**) and to monitor animal movement (**e**: motion index). **a-b**): time-lapse images ratio changes $\Delta\mu\text{ODT}$ (%) in original **Fig.7b** and their 3D $\mu\text{ODT}(t)$ images to illustrate motion artifacts at cross sections of $i=1$ to 10 without AI denoising, image size: $2.25 \times 0.3 \times 1.2 \text{mm}^3$; **c-d**): the corresponding images of (**a-b**) after AI denoising to effectively remove motion artifacts of animal movements - new **Fig.7b**. **e**) Motion index (i.e., decorrelation) to illustrate the timepoints or B-scans ($i=1, \dots, 10$) of animal movement severity.

These are good questions. We previously reported TIA and the resultant motor dysfunction (e.g., hemiparalysis, paralysis) elicited by chronic cocaine, but the observation of TIA was under isoflurane anesthesia, in which anesthetic confounds could not be ruled out. This is one of the major reasons we develop AI-enhanced μ OCA/ μ ODT for awake animal studies. As discussed above, a limitation of this study is that neural activity was not measured, so we could not characterize the differences in sensory- and motor-related neural activity between awake and anesthetized conditions. Therefore, we cannot elaborate on the contribution of the hemodynamic signals to specific neuronal activities, but we are developing and validating a hybrid imaging system that is tailored for separating these changes. This limitation is now discussed in the revised manuscript (pg 24, para 2).

Minor:

5. "Fig. 3c-d – colored bar segments should be more visible – perhaps make them thicker?"

Thanks, and we have changed them accordingly.

6. "Fig.3e – this is a bit confusing – all of the data points should be shown, i.e. the baseline condition and then the iso condition. Also - the label delta phi should be labeled something more intuitive – what is this metric? How does it apply to CD? Also – it doesn't make visual sense nor communicate much data to the reader to see a ~0 delta in CD; this is where showing the actual baseline vs. iso condition raw data points could help. Additionally, it should be described in method show capillary density was calculated."

We agree and have changed them accordingly. To avoid confusion, we separate arterial/venal vessel diameter changes (left y-axis, $\Delta\phi/\phi$) and capillary density change (right y-axis, $\Delta D/D$). We have also included the method for capillary density calculation, which includes segmentation, skeletonization and fill factor calculation. Please see changes in pg 28, para 3, Supplementary **S5** and the links to our previous papers for details.

7. "Fig. 5e – same issues as Fig. 3e. Here, there is basically no information conveyed by the plot. It's not obvious to me what the small numbers punctuating the traces in Figs. 4f, 6f, 7c represent. This should be stated in the figure captions."

Fig.3e and **Fig.5e** were included to indicate the resultant vascular changes, including vasodilation and vasoconstriction in arterial and venular vessels ($\Delta\phi/\phi$) and the capillary density changes ($\Delta D/D$). To avoid confusion, we have included a right axis (y-axis) to clarify the capillary density changes.

We now increase the font size and use red, light blue and green fonts to separate arterial, venular and capillary flows to be consistent with the color of the traces. We also add figure captions to clarify the differences. Please see changes in the corresponding figures and captions.

https://www.cdc.gov/nchs/pressroom/nchs_press_releases/2021/20211117.htm

Boag, F., and Havarad, C.W. (1985). Cardiac arrhythmia and myocardial ischaemia related to cocaine and alcohol consumption. *Postgrad Med J* *61*, 997-999.

Bortel, A., Pilgram, R., Yao, Z.S., and Shmuel, A. (2020). Dexmedetomidine - Commonly Used in Functional Imaging Studies - Increases Susceptibility to Seizures in Rats But Not in Wild Type Mice. *Front Neurosci* *14*, 832.

Bryant, J.E., Frolich, M., Tran, S., Reid, M.A., Lahti, A.C., and Kraguljac, N.V. (2019). Ketamine induced changes in regional cerebral blood flow, interregional connectivity patterns, and glutamate metabolism. *J Psychiatr Res* *117*, 108-115.

Busse, G.D., and Riley, A.L. (2003). Effects of alcohol on cocaine lethality in rats: acute and chronic assessments. *Neurotoxicol Teratol* *25*, 361-364.

Franks, N.P., and Lieb, W.R. (1994). Molecular and cellular mechanisms of general anaesthesia. *Nature* *367*, 607-614.

Goldstein, D.B. (1984). The effects of drugs on membrane fluidity. *Annu Rev Pharmacol Toxicol* *24*, 43-64.

Gu, X., Chen, W., Volkow, N.D., Koretsky, A.P., Du, C., and Pan, Y. (2018a). Synchronized Astrocytic Ca(2+) Responses in Neurovascular Coupling during Somatosensory Stimulation and for the Resting State. *Cell Rep* *23*, 3878-3890.

Gu, X., Chen, W., You, J., Koretsky, A.P., Volkow, N.D., Pan, Y., and Du, C. (2018b). Long-term optical imaging of neurovascular coupling in mouse cortex using GCaMP6f and intrinsic hemodynamic signals. *Neuroimage* *165*, 251-264.

Jelescu, I.O., Ciobanu, L., Geffroy, F., Marquet, P., and Le Bihan, D. (2014). Effects of hypotonic stress and ouabain on the apparent diffusion coefficient of water at cellular and tissue levels in Aplysia. *NMR Biomed* *27*, 280-290.

Keilholz, S.D., Silva, A.C., Raman, M., Merkle, H., and Koretsky, A.P. (2004). Functional MRI of the rodent somatosensory pathway using multislice echo planar imaging. *Magn Reson Med* *52*, 89-99.

Magnuson, M.E., Thompson, G.J., Pan, W.J., and Keilholz, S.D. (2014). Time-dependent effects of isoflurane and dexmedetomidine on functional connectivity, spectral characteristics, and spatial distribution of spontaneous BOLD fluctuations. *NMR Biomed* *27*, 291-303.

Montijn, J.S., Meijer, G.T., Lansink, C.S., and Pennartz, C.M. (2016). Population-Level Neural Codes Are Robust to Single-Neuron Variability from a Multidimensional Coding Perspective. *Cell Rep* *16*, 2486-2498.

Nelson, L.E., Lu, J., Guo, T., Saper, C.B., Franks, N.P., and Maze, M. (2003). The alpha2-adrenoceptor agonist dexmedetomidine converges on an endogenous sleep-promoting pathway to exert its sedative effects. *Anesthesiology* *98*, 428-436.

Park, K., Chen, W., Volkow, N.D., Allen, C.P., Pan, Y., and Du, C. (2019). Hemodynamic and neuronal responses to cocaine differ in awake versus anesthetized animals: Optical brain imaging study. *Neuroimage* *188*, 188-197.

Park, K., Liyanage, A.C., Koretsky, A.P., Pan, Y., and Du, C. (2021). Optical imaging of stimulation-evoked cortical activity using GCaMP6f and jRGECO1a. *Quant Imaging Med Surg* *11*, 998-1009.

Pawela, C.P., Biswal, B.B., Hudetz, A.G., Schulte, M.L., Li, R., Jones, S.R., Cho, Y.R., Matloub, H.S., and Hyde, J.S. (2009). A protocol for use of medetomidine anesthesia in rats for extended studies using task-induced BOLD contrast and resting-state functional connectivity. *Neuroimage* *46*, 1137-1147.

Rehberg, B., Xiao, Y.H., and Duch, D.S. (1996). Central nervous system sodium channels are significantly suppressed at clinical concentrations of volatile anesthetics. *Anesthesiology* *84*, 1223-1233; discussion 1227A.

Simmler, L.D., Li, Y., Hadjas, L.C., Hiver, A., van Zessen, R., and Luscher, C. (2022). Dual action of ketamine confines addiction liability. *Nature* *608*, 368-373.

Torres, G., Rivier, C., and Weiss, F. (1994). A ketamine mixture anesthetic inhibits neuroendocrine and behavioral consequences of cocaine administration. *Brain Res* *656*, 33-42.

Tsukada, H., Nishiyama, S., Kakiuchi, T., Ohba, H., Sato, K., and Harada, N. (2001). Ketamine alters the availability of striatal dopamine transporter as measured by [(11)C]beta-CFT and [(11)C]beta-CIT-FE in the monkey brain. *Synapse* *42*, 273-280.

Weber, R., Ramos-Cabrera, P., Wiedermann, D., van Camp, N., and Hoehn, M. (2006). A fully noninvasive and robust experimental protocol for longitudinal fMRI studies in the rat. *Neuroimage* *29*, 1303-1310.

Zanos, P., Moaddel, R., Morris, P.J., Riggs, L.M., Highland, J.N., Georgiou, P., Pereira, E.F.R., Albuquerque, E.X., Thomas, C.J., Zarate, C.A., Jr., *et al.* (2018). Ketamine and Ketamine Metabolite Pharmacology: Insights into Therapeutic Mechanisms. *Pharmacol Rev* *70*, 621-660.

J You, A Li, C Du, & Y Pan, "Volumetric Doppler angle correction for ultrahigh-resolution optical coherence Doppler tomography," *APPLIED PHYSICS LETTERS* *110*, 011102 (2017).

H Ren, C Du, Z Yuan, K Park, ND Volkow and Y Pan, "Cocaine-induced cortical microischemia in the rodent brain: clinical implications," *Mol Psychiatry*, 2012 *17*(10):1017-25. doi: 10.1038/mp.2011.160.

Ang Li, Jiang You, Congwu Du, and Yingtian Pan, "Automated segmentation and quantification of OCT angiography

for tracking angiogenesis progression," *Biomed Opt Express*. 2017 8(12): 5604–5616. doi:10.1364/BOE.8.005604

Li, A., Du, C., Volkow, N. D. & Pan, Y. A deep-learning-based approach for noise reduction in high-speed optical coherence Doppler tomography. *J Biophotonics* 13, e202000084, doi:10.1002/jbio.202000084 (2020).

41 Li, A., Du, C. & Pan, Y. Deep-learning-based motion correction in optical coherence tomography angiography. *J Biophotonics* 14, e202100097, doi:10.1002/jbio.202100097 (2021).

You, J. et al. Cerebrovascular adaptations to cocaine-induced transient ischemic attacks in the rodent brain. *JCI Insight* 2, e90809, doi:10.1172/jci.insight.90809 (2017)

Rosenbaum SB, et al. Ketamine. [Updated 2022 Nov 24]. In: StatPearls [Internet]. Treasure Island (FL): StatPearls Publishing; 2022 Jan-. Available from: <https://www.ncbi.nlm.nih.gov/books/NBK470357/>

Lamont LA, et al., Physiology of pain. *Vet Clin North Am Small Anim Pract*. 2000 30(4):703–28, v. doi: 10.1016/s0195-5616(08)70003-2. PMID: 10932821.

Veilleux-Lemieux D, et al., Pharmacokinetics of ketamine and xylazine in young and old Sprague-Dawley rats. *J Am Assoc Lab Anim Sci*. 2013 52(5):567–70. PMID: 24041212; PMCID: PMC3784662.

REVIEWERS' COMMENTS:

Reviewer #1 (Remarks to the Author):

The authors have done an excellent job in revising their manuscript and have addressed nearly all of my concerns with the addition of several new figures and analyses. A remaining, but relatively minor, concern is the lack of motivation for examining cocaine in the title and abstract. The authors are encouraged to frame their study in such a way that might attract a broader audience to their work. This comment does not require a re-review from me.

Reviewer #3 (Remarks to the Author):

The authors have done a good job of responding to my concerns.

Manuscript tracking #: COMMSBIO-22-1480-T

Title: "Dynamic 3D imaging of cerebral blood flow networks in awake mice using ultrahigh-resolution optical coherence Doppler tomography"

Response to the Referee Comments

We thank the reviewers for their comments and suggestions on the manuscript and have revised the manuscript accordingly. The revisions are marked in yellow highlights in the revised manuscript and the revised Supplementary Information. In addition, the corresponding clean versions are provided for final consideration.

Reviewer #1:

"The authors have done an excellent job in revising their manuscript and have addressed nearly all of my concerns with the addition of several new figures and analyses. A remaining, but relatively minor, concern is the lack of motivation for examining cocaine in the title and abstract. The authors are encouraged to frame their study in such a way that might attract a broader audience to their work. This comment does not require a re-review from me."

We agree and have revised the title and the abstract accordingly.

We agree on the editor's brief summary in the Final Revision Instruction (page 3) and would like to change the title as "Ultrahigh-resolution optical coherence Doppler tomography with self-supervised deep learning allows for 3D imaging of cerebral blood flow changes in response to cocaine in awake mice". Because of limited word counts, we have included a statement "a highly addictive drug associated with neurovascular toxicity" in the Abstract to justify the use of cocaine.

Please see the changes in yellow highlighted areas on page 1 and page 2.

Editor's Final Revision Instructions:

1. *"Each Figure must be provided as a separate file"*

We have saved all 9 figures in separate files. Each figure is saved in tiff format with a resolution of 300dpi. Besides, we have moved the figure captions to the end of the manuscript.

2. *"Your paper will be accompanied by a brief editor's summary when it is published on our homepage. Please approve the draft summary below or provide us with a suitably edited version (no more than 250 characters including spaces)."*

We like the summary: An imaging platform with self-supervised deep learning allows for the imaging of cerebral blood flows under the effect of cocaine in awake mice using 3D ultrahigh-resolution optical coherence Doppler tomography, in which ultrahigh-resolution that we include is optional.

3. *Changes regarding Supplementary Information including format, referencing, naming, movie files, numerical data for graphs and charts, ... (Final Revision Instructions, page 5-6)*

We have carefully revised them following the guidelines and marked up these changes in yellow highlights.

4. *"We recommend that author first names be written out in full rather than provided as initials.*

At least one corresponding author must be designated, and an e-mail address must be provided for each corresponding author (with a limit of one e-mail address per author"

We have provided full names for all the authors and the email of the corresponding author.

5. *"The editors recommend the following title: Dynamic 3D imaging of cerebral blood flow in awake mice using self-supervised-learning-enhanced optical coherence Doppler tomography"*

We like the editor's title "Dynamic 3D imaging of cerebral blood flow in awake mice using self-supervised-learning-enhanced optical coherence Doppler tomography".

6. *"Abstract: Final sentence indicating any broader impacts and how this research will be used in the future. The editors recommend the following edits to your abstract: Please add a broader impact sentence and consider removing the statistical details from the abstract."*

We thank the editor's comments and suggestions. We now remove all the statistical details and add a final statement to indicate the broader impacts:

"The 3D imaging platform we present provides a powerful tool to study dynamic changes in vessel diameters and morphology alongside CBFv networks in the brain of awake animals that can advance our understanding of the effects of drugs and disease conditions (ischemia, tumors, wound healing)."

7. *"Avoid the use of the word "significant" unless referring the results of a statistical test."*

We agree and have removed all the statements involving significant but without p-value.

8. *"Figures: Abbreviations, symbols, colors, and shading present in the Figure must be defined. Please write out the symbols/colors in words (blue circles, red dashed line, etc.) within these definitions. you must add individual data points or convert the graph to a boxplot or dot-plot."*

We have defined abbreviations in the figure captions and revised the plots to include data points in the new Figures 1-9 and Supplementary figures.

9. *"Please add a Data Availability statement."*

We have included Data Availability Statement on page 24, included all the numerical data files for the graphs and plots, and provided web links to programs and codes. The original 3D image datasets are giant (>10GB per image), so we include a statement 'available upon request'.

10. *"Please provide 'Competing interests' and 'Author Contributions' sections."*

We have included a section to declare no competing interests and a section to list authors' contributions.